



# Advanced methods for uncertainty assessment and global sensitivity analysis of a Eulerian atmospheric chemistry transport model

Ksenia Aleksankina,[1,2] Stefan Reis[2,3], Massimo Vieno[2], and Mathew R. Heal[1]

[1] School of Chemistry, University of Edinburgh, Edinburgh, UK

[2] NERC Centre for Ecology & Hydrology, Penicuik, UK

[3] University of Exeter Medical School, European Centre for Environment and Health, Knowledge Spa, Truro, UK

*Correspondence to*: Ksenia Aleksankina (k.aleksankina@sms.ed.ac.uk) and Mathew Heal (m.heal@ed.ac.uk)

**Abstract.** Atmospheric chemistry transport models (ACTMs) are extensively used to provide scientific support for the development of policies to mitigate against the detrimental effects of air pollution on human health and ecosystems. Therefore,

it is essential to quantitatively assess the level of model uncertainty and to identify the model input parameters that contribute the most to the uncertainty. For complex process-based models, such as ACTMs, uncertainty and global sensitivity analyses are still challenging and are often limited by computational constraints due to the requirement of a large number of model runs. In this work, we demonstrate an emulator-based approach to uncertainty quantification and variance-based sensitivity analysis for the EMEP4UK model (regional application of the European Monitoring and Evaluation Programme Meteorological

Synthesizing Centre-West). A separate Gaussian process emulator was used to estimate model predictions at unsampled points in the space of the uncertain model inputs for every modelled grid cell. The training points for the emulator were chosen using an optimised Latin hypercube sampling design. The uncertainties in surface concentrations of $O_3$, $NO_2$, and $PM_{2.5}$ were propagated from the uncertainties in the anthropogenic emissions of $NO_x$, $SO_2$, $NH_3$, VOC, and primary $PM_{2.5}$ reported by the UK National Atmospheric Emissions Inventory. The results of the EMEP4UK uncertainty analysis for the annually averaged

model predictions indicate that modelled surface concentrations of $O_3$, $NO_2$, and $PM_{2.5}$ have the highest level of uncertainty in the grid cells comprising urban areas (up to $\pm 7\%$, $\pm 9\%$, and $\pm 9\%$ respectively). The uncertainty in the surface concentrations of $O_3$ and $NO_2$ were dominated by uncertainties in $NO_x$ emissions combined from non-dominant sectors (i.e. all sectors excluding energy production and road transport) and shipping emissions. Additionally, uncertainty in $O_3$ was driven by uncertainty VOC emissions combined from sectors excluding solvent use. Uncertainties in the modelled $PM_{2.5}$ concentrations

were mainly driven by uncertainties in primary $PM_{2.5}$ emissions and $NH_3$ emissions from the agricultural sector. Uncertainty and sensitivity analyses were also performed for five selected grid sells for monthly averaged model predictions to illustrate the seasonal change in the magnitude of uncertainty and change in the contribution of different model inputs to the overall uncertainty. Our study demonstrates the viability of a Gaussian process emulator-based approach for uncertainty and global sensitivity analyses, which can be applied to other ACTMs. Conducting these analyses helps to increase the confidence in

model predictions. Additionally, the emulators created for these analyses can be used to predict the ACTM response for any





other combination of perturbed input emissions within the ranges set for the original Latin hypercube sampling design without the need to re-run the ACTM, thus allowing fast exploratory assessments at significantly reduced computational costs.

## 1 Introduction

Air pollution has a wide range of detrimental impacts. Exposure to air pollutants such as nitrogen dioxide ($NO_2$), ozone ($O_3$),
and particulate matter ($PM_{2.5}$) is associated with increased risk of stroke, cardiovascular disease, and chronic and acute respiratory diseases (WHO, 2006, 2013). Additionally, particulate matter and $O_3$ contribute to climate change through radiative forcing (IPCC, 2013; Stevenson et al., 2013) and $O_3$ has an adverse impact on natural and semi-natural vegetation and crop yields (Teixeira et al., 2011).

To reduce the harmful impact of air pollution, various policies and directives have been implemented. For example, in the
European Union, the Ambient Air Quality Directive (EC Directive, 2008) sets limit values on ambient concentrations of air pollutants, whilst other directives set source-specific emissions limits. Atmospheric Chemistry Transport Models (ACTMs) play an essential role in the evaluation of the potential outcomes of different management options aimed at improvement of future air quality.

The majority of existing ACTMs are deterministic, meaning that the output variables are presented as a single value without
any indication of the expected uncertainty around this value. The uncertainty estimate for the modelled value is critical because it provides an assessment of confidence in the model predictions and the confidence range may encompass different recommendations that can be drawn from the model (Frost et al., 2013; Rypdal and Winiwarter, 2001). It has been previously found that uncertainties in input emissions are major contributors to the uncertainty in the ACTM outputs (Hanna et al., 2007; Rodriguez et al., 2007; Sax and Isakov, 2003). Therefore, this study concentrates on implementing a systematic approach for
ACTM output uncertainty quantification and on determining the extent to which different input emissions drive the uncertainty in the output variables.

Analytical uncertainty propagation is not feasible for complex models such as ACTMs because it requires an exact function for input-output mapping. Consequently, Monte Carlo based methods for uncertainty assessment have to be used. Uncertainty analysis should be performed in tandem with sensitivity analysis to maximise the knowledge gained. Sensitivity analysis
provides information on how the overall uncertainty can be apportioned to different model inputs and hence allows conclusions to be drawn on the extent to which the overall variation in the modelled values is driven by variation in different inputs (Saltelli, 2002).

For computationally demanding models, such as ACTMs, a local one-at-a-time (OAT) sensitivity analysis is the most commonly used approach (Ferretti et al., 2015). However, unlike global sensitivity analysis, the local OAT approach does not
take into account the non-linearities in the model response and the interactions between the input parameters (Aleksankina et al., 2018; Saltelli and Annoni, 2010).



The computational cost of running ACTMs to explore the entire parameter space of the uncertain inputs using Monte Carlo based uncertainty and sensitivity analyses is typically prohibitively high because the analyses require a large number of points in parameter space which translates to thousands of model simulations. To tackle this issue, the use of meta-models has been increasing in recent years (Gladish et al., 2017; Iooss and Lemaître, 2015; Ratto et al., 2012; Yang, 2011). A meta-model (or

emulator) is a statistical approximation of the original simulation model that can be evaluated many times at a lower computational cost relative to the original model (Castelletti et al., 2012; O'Hagan, 2006). This approach allows the output of an ACTM for a large number of points in parameter space to be estimated efficiently making uncertainty and sensitivity analyses feasible.

Different meta-modelling approaches have been used for uncertainty and sensitivity analysis; these techniques include

regression smoothers (Storlie et al., 2009; Storlie and Helton, 2008), Gaussian process emulator (Oakley and O'Hagan, 2004), high-dimensional model representation (Rabitz and Alış, 1999; Ziehn and Tomlin, 2009), and polynomial chaos expansion (Sudret, 2008). Meta-models have been applied for uncertainty and sensitivity analyses in earth science fields such as ecological modelling (Luo et al., 2013; Parry et al., 2013), hydrological modelling (Asher et al., 2015; Gladish et al., 2017), and atmospheric aerosol modelling (Carslaw et al., 2013; Chen et al., 2013; Christian et al., 2017; Lee et al., 2011).

In this study, a Gaussian process is used for emulation because of its desirable properties and available implementations (i.e. Matlab based software UQLab or R package DiceKriging). Gaussian process emulators are non-parametric statistical models that use the principles of conditional probability to estimate model outputs. The beneficial properties are the curve that fits through the training points (for deterministic models) and a measure of the uncertainty for the estimated points when using an emulator in place of the original model for the estimation of new points.

The efficiency of the emulator compared to the original model is determined by how smooth and continuous the model response is to input perturbations. For a smooth and continuous input-output relationship, the high correlation between the inputs and the simulated points means a lower uncertainty in predictions made using the emulator further away from the training points (i.e. resulting in a good emulator performance with a small number of training points) (Lee et al., 2011).

The design of computer experiments for deterministic models differs from the designs for physical experiments. As there is

no random error involved in computer experiments, replication is not required (Jones and Johnson, 2009). Hence sampling techniques that have good space-filling properties and the ability to maintain uniform spacing when projected into a lower-dimensional space are used (Dean et al., 2015; Jones and Johnson, 2009). Latin hypercube sampling (LHS) introduced by (McKay et al., 1979) meets these desirable criteria. Additionally, advances have been made to optimise the space filling properties of LHS including maximin sampling (Johnson et al., 1990; Morris and Mitchell, 1995) and the ability to add extra

design points to the parameter space if necessary (Sheikholeslami and Razavi, 2017) which makes it well suited for multi-dimensional designs that may require the addition of extra points.

The aim of this study is to demonstrate the method for uncertainty assessment and global sensitivity analysis for computationally demanding ACTMs. The ACTM to which the method is applied here is the WRF-EMEP4UK model (Vieno et al., 2010, 2014, 2016a), and the outputs of interest are  the modelled surface concentrations of $O_3$, $NO_2$, and $PM_{2.5}$, but the



methodology is generic for model and output variable. The analyses described here investigated sensitivities and uncertainties of model output to emissions from UK land-based sources and from surrounding shipping. Additionally, we identify which model inputs drive uncertainty in the output variables, and to what extent; as well as discuss how the uncertainty ranges that are obtained affect current predictions/scenario analysis outcomes (i.e. confidence in model outputs).

## 2 Methods

### 2.1 Model description

The EMEP4UK model is a regional application of the EMEP MSC-W (European Monitoring and Evaluation Programme Meteorological Synthesizing Centre-West) open source ACTM (www.github.com/metno/emep-ctm, version rv4.8, last access: 11 June 2018). The detailed description of EMEP MSC-W is available from Simpson et al. (2012), and the EMEP4UK model is described by (Vieno et al., 2010, 2014, 2016a).

EMEP4UK is a 3-D one-way nested Eulerian model with a horizontal resolution of 5 km × 5 km over the British Isles nested within an extended European domain with 50 km × 50 km resolution. The extent of the inner domain is shown in Figure 1. The model has 20 vertical levels, extending from the ground to 100 hPa with the lowest vertical layer of ~90 m. The model time-step is 20 s for chemistry, 5 min for the advection in the inner domain, and 20 min for the advection in the outer domain.

The meteorological fields were computed using Weather Research and Forecast model version 3.1.1 (www.wrf-model.org, last access: 15 November 2017) (Skamarock et al., 2008). The WRF model initial and boundary conditions are derived from the US National Center for Environmental Prediction (NCEP)/National Center for Atmospheric Research (NCAR) Global Forecast System (GFS) at 1∘ resolution, including Newtonian nudging every 6 h (NCEP, 2000).

The anthropogenic emissions of $SO_2$, $NO_x$, $NH_3$, primary $PM_{2.5}$, primary $PM_{coarse}$, CO, and non-methane VOC for the UK were derived from the National Atmospheric Emissions Inventory (http://naei.beis.gov.uk/, last access: 15 October 2015). For the outer domain, the emissions are provided by the Centre for Emission Inventories and Projections (CEIP, http://www.ceip.at/, last access: 15 October 2017). All emissions are split across a set of emission source sectors defined by the Selected Nomenclature for Air Pollutants (SNAP) described in Table 1. The hour-of-day, day-of-week and monthly emission factors are used to distribute the annual total emissions temporally to hourly resolution as described in Simpson et al. (2012). The international shipping emissions were derived from ENTEC UK Ltd. (now Amec Foster Wheeler). Biogenic emissions of dimethyl sulfide in addition to monthly in-flight aircraft, soil, and lightning $NO_x$ emissions are included as described in Simpson et al. (2012). Biogenic emissions of monoterpenes and isoprene are calculated by the model for every grid cell and time step. The emissions of sea salt and wind-blown dust are also included.

The chemistry, aerosol formation, and wet and dry deposition schemes are as described in Simpson et al. (2012). The chemistry scheme has 72 species, 137 reactions, and the gas/aerosol partitioning is described by the MARS formulation. A detailed evaluation of model performance is discussed elsewhere (Dore et al., 2015; Lin et al., 2017; Vieno et al., 2010, 2016b). In our study, all model runs were executed using meteorology and emissions data for the year 2012.



### 2.2 Input variables and their uncertainty ranges

For this study, emissions of five pollutants ($NO_x$, $SO_x$, VOC, $NH_3$, primary $PM_{2.5}$) were split into 13 model input variables based on the contributions from different emission source sectors to total annual emissions; the emissions from the dominant sector (the sector with the highest relative contribution to total emissions) for every pollutant were treated as a separate variable, while the emissions from the rest of the sectors were grouped and treated as another input variable. Shipping emissions were treated as a separate variable and were not split by the pollutant type. The description of the Selected Nomenclature for Air Pollution (SNAP) sectors is shown in Table 1, and the definitions of the input variables for the uncertainty and sensitivity analyses in this work are presented in Table 2, where variables marked with D represent emissions from a single dominant sector (D1 and D2 in case of multiple dominant sectors) and variables marked with O indicate the grouped 'other' emissions from the rest of the sectors. Emissions from 'natural' sources (e.g. lightening, soil, ocean) were not part of the uncertainty and sensitivity analyses.

Uncertainty ranges for the input emissions from UK anthropogenic land-based sources were assigned according to data in the UK Informative Inventory Report (IIR) (Wakeling et al., 2017). In the IIR, uncertainties are defined as upper and lower limits of the 95% confidence interval relative to the central estimate. There is no information on uncertainty ranges for different source sectors available for the emissions for 2012 because uncertainties split by the emission source sector were first presented in the IIR that included 2014 emissions (Wakeling et al., 2016). Hence, for this study, the most recently published data for the uncertainty ranges of pollutants split by source sector were used.

Equation 1 was used to aggregate uncertainties for multiple emission source sectors for the grouped-source input variables, where $x$ is the quantity of interest and $U$ is the uncertainty of that quantity, taken from the EMEP/EEA air pollutant emission inventory guidebook (Pulles and Kuenen, 2016).

$$U_{total} = \frac{\sqrt{(U_1\,x_1)^2 + (U_2\,x_2)^2 + \cdots + (U_n\,x_n)^2}}{x_1 + x_2 + \cdots + x_n} \tag{1}$$

The shipping emission variable in this study combines all emissions of all relevant pollutants, hence a 'best estimate' range for the uncertainty was chosen. The range was estimated based on the available published information. Some recently published sources (Corbett, 2003; Scarbrough et al., 2017) state that the uncertainty in shipping emissions is significant, but do not provide quantitative estimates. The most recent source of quantitative information on the uncertainty in shipping emissions is the report for the European Commission (Entec, 2002) which presents the estimates of uncertainties for emission factors of $NO_x$, $SO_2$, PM, VOC, for the ships' emissions 'at sea', 'manoeuvring', and 'in port'. The uncertainties are presented for the emissions for the year 2000 as 95% CI with the lowest values of uncertainty presented for 'at sea' emission factors ($\pm$ 10-20%) and highest values for 'manoeuvring' emission factors ($\pm$ 30-50%). For the total pollutant emissions for the year 2000 the percentage uncertainties around the estimates are $\pm$ 21% for $NO_x$, $\pm$ 11% for $SO_2$, $\pm$ 11% for $CO_2$, $\pm$ 28% for VOC and $\pm$45% for PM. Additionally, in Moreno-Gutiérrez et al. (2015) the uncertainty in the emission factors for all pollutant compounds



was estimated to be ± 20%. Using the above data, an overall uncertainty of ± 30% was assigned to the shipping emissions variable in this study (Table 2). It was applied to all shipping emissions within the inner British Isles domain of the EMEP4UK model.

## 2.3 Gaussian process emulator for EMEP4UK

A Gaussian Process emulator was used to estimate model predictions at unsampled points in the space of the uncertain model inputs. The UQLab, a MATLAB-based software framework for uncertainty quantification (Lataniotis et al., 2017; Marelli and Sudret, 2014), was implemented to build the emulators for the uncertainty propagation and the following sensitivity analysis. The comprehensive description of the statistical theory of Gaussian process applied to uncertainty and sensitivity analysis with full mathematical details can be found in O'Hagan (2006) and Oakley and O'Hagan (2002, 2004).

The uncertainty values and sensitivity indices were calculated for three EMEP4UK model outputs ($O_3$, $NO_2$, and $PM_{2.5}$ surface concentrations) with annual and monthly temporal resolution. For the annually-averaged outputs, an emulator was created for each modelled grid cell in the EMEP4UK domain ($n = 59\ 400$). The first and total-order sensitivity indices were calculated for the land-based grid cells only ($n > 10\ 000$). For the monthly mean model outputs, uncertainty and sensitivity analysis were performed for five selected grid cells. The five grid cells were selected to contain a UK national-network air pollution

monitoring station to aid classification according to the environment (i.e. rural background, urban background, and urban traffic) and also to provide geographically representative coverage across the UK.

*LHS maximin* design, which maximises the minimum distance between the points in the parameter space to provide the optimum space-filling properties was used. The design was previously demonstrated suitable for Gaussian process emulators by Jones and Johnson (2009). The design with 84 data points was created for the scaling coefficients that were subsequently

applied to the input emissions. This means that emissions corresponding to a particular input variable were perturbed homogeneously throughout the whole of the UK model domain. The ranges of scaling coefficient used for the sampling design are presented in Table 2.

In this study, the surface concentration of $O_3$, $NO_2$, and $PM_{2.5}$ for every grid cell is defined as a scalar output $Y = f(X)$ where $X$ is the vector of input values $\{X_1, \dots, X_{13}\}$.

A Gaussian process emulator utilises a Bayesian approach; the training data is used to update the selected prior to produce posterior mean and covariance functions. The Gaussian process is specified by its mean function and covariance function. The mean function is given by Eq. 2:

$$\mathbb{E}[f(x)|\boldsymbol{\beta}] = \boldsymbol{h(x)}^T \boldsymbol{\beta} \qquad (2)$$

where $\boldsymbol{h(\cdot)}$ is a vector of regression functions and $\boldsymbol{\beta}$ is a vector of unknown coefficients. The choice of $\boldsymbol{h(\cdot)}$ incorporates any prior beliefs about the form of $f(\cdot)$. In this study, the mean function was chosen to have a linear form $\beta_o + \sum_{i=1}^{13} \beta_i x_i$.





The covariance function between $f(\boldsymbol{x})$ and $f(\boldsymbol{x'})$ is given by Eq. 3:

$$cov\{f(\boldsymbol{x}), f(\boldsymbol{x'})|\sigma^2\} = \sigma^2 c(\boldsymbol{x}, \boldsymbol{x'}) \qquad (3)$$

where $\sigma^2$ is the hyperparameter that represents the variance of the Gaussian process and $c(\boldsymbol{x}, \boldsymbol{x'})$ is the correlation function.
The correlation function increases as the distance between $\boldsymbol{x}$ and $\boldsymbol{x'}$ decreases and equals one when $\boldsymbol{x} = \boldsymbol{x'}$. In this study Matérn 5/2 (Eq. 4) was used, where $h$ is the absolute distance between $\boldsymbol{x}$ and $\boldsymbol{x'}$ and $\boldsymbol{\theta}$ is a vector of range parameters or length-scales, which define how far one needs to move along a particular axis in the input space for the function values to become uncorrelated.

$$c(x, x') = \left(1 + \frac{\sqrt{5}|h|}{\theta} + \frac{5\,h^2}{3\theta^2}\right) \exp\left(-\frac{\sqrt{5}|h|}{\theta}\right) \qquad (4)$$

A number of emulators were built with the EMEP4UK simulation data using other available covariance functions; however, little difference was found in the performance of the emulators. The hyperparameters $\boldsymbol{\beta}$, $\sigma^2$, and $\boldsymbol{\theta}$ were estimated using a cross-validation approach.

The emulator error was estimated by implementing $k$-fold cross-validation (Gladish et al., 2017; Urban and Fricker, 2010). The original sample was randomly partitioned into $k = 10$ sized subsamples which allowed approximately 90% of data to be used as a training set and 10% as a validation set. Spatial distribution of cross-validation errors is presented in the supplementary information (Figure S1).

## 2.4 Uncertainty and sensitivity analysis

### 2.4.1 Uncertainty propagation

The uncertainties for the EMEP4UK output variables were estimated using a Monte Carlo approach (also described in the IPCC guidelines (IPCC, 2006) as a Tier 2 approach). The specific uncertainty ranges assigned to the input emission variables were used to constrain the input sampling space. All inputs were assigned normal distributions with baseline value as the mean and the standard deviation derived from the corresponding confidence interval (Table 1). For every grid cell, the emulator was used to predict model values of surface concentrations of $O_3$, $NO_2$, and $PM_{2.5}$ at the new set of input points ($n = 5,000$). The resulting probability distributions for each grid cell were evaluated, and the resulting uncertainty was estimated as a half of the 95% confidence interval relative to the central estimate (i.e. the mean for normally distributed values) of the output value, as described in the EMEP/EEA and IPCC Guidebooks (IPCC, 2006; Pulles and Kuenen, 2016). The uncertainty for the monthly average modelled surface concentrations of $O_3$, $NO_2$, and $PM_{2.5}$ was calculated for five grid cells using the same approach as above. The locations of the grid cells within the UK are shown in Figure 1. The five grid cells selected were assigned the following environment types – the names and environment type reflect those of the national-network monitoring site within



that grid cell: Auchencorth Moss and Harwell - rural background, Birmingham Acocks Green and London N. Kensington - urban background, and London Marylebone Road - urban traffic.

**2.4.2 Global sensitivity analysis; first- and total-order indices**

A variance-based global sensitivity analysis was conducted to apportion overall uncertainty in modelled variables to the uncertainty in the input emissions. Sobol' first and total-order sensitivity indices were estimated (Homma and Saltelli, 1996; Janon et al., 2014; Sobol', 2001, 1993). The first-order indices represent the fraction of total variance of the output (i.e. the proportion of the overall uncertainty in $Y$) explained by the variance in an input $X_i$ while total-order indices show the sum of the effects due to an input $X_i$ and all of its interactions with other inputs ($X_{\sim i}$). Therefore, the values of first and total-order indices can be compared to identify the presence of interactions between input $X_i$ and all other model inputs.

Unlike an OAT sensitivity coefficient, a first-order sensitivity index accounts for the non-linear response of a model output to a parameter across the specified parameter variation range. Sensitivity indices in this context are also indicators of importance for the input variables.

The first-order sensitivity index is defined as the ratio of the variance of the mean of $Y$ when one input variable is fixed, $V_{Xi}(E_{X\sim i}(Y/X_i))$, to the unconditional variance of $Y$, $V(Y)$ (Eq. 5).

$$S_i = \frac{V_{X_i}(E_{\boldsymbol{X}_{\sim i}}(Y|X_i))}{V(Y)} \tag{5}$$

The total order sensitivity index measures the total effect of a variable, which includes its first-order effect and interactions with any other variables (Eq. 6).

$$S_{Ti} = 1 - \frac{V_{\boldsymbol{X}_{\sim i}}\left(E_{X_i}(Y|\boldsymbol{X}_{\sim i})\right)}{V(Y)} = \frac{E_{\boldsymbol{X}_{\sim i}}\left(V_{X_i}(Y|\boldsymbol{X}_{\sim i})\right)}{V(Y)} \tag{6}$$

where $\boldsymbol{X}_{\sim i}$ denotes the matrix of all variables but $X_i$. In $E_{X\sim i}(V_{Xi}(Y/\boldsymbol{X}_{\sim i}))$ the inner variance of $Y$ is taken over all possible values of $X_i$ while keeping $\boldsymbol{X}_{\sim i}$ fixed, while the output expectation $E$ is taken over all possible values $\boldsymbol{X}_{\sim i}$ (Ghanem et al., 2017). For the annual average modelled surface concentrations of $O_3$, $NO_2$ and $PM_{2.5}$, the sensitivity indices were calculated for the UK land-based grid cells for the whole domain. For the monthly average modelled concentrations, sensitivity indices for five selected grid cells (discussed above) were estimated to determine whether seasonality affects the magnitude of the sensitivity indices.





## 3 Results and discussion

### 3.1 Uncertainty propagation

Figure 2 shows the spatial distribution of annual average surface concentrations of $O_3$, $NO_2$, and $PM_{2.5}$ modelled with EMEP4UK and their absolute and relative uncertainties given the uncertainties in UK pollutant emissions for each source
sector shown in Table 2. The uncertainties are presented as a range of ± the baseline value and represent the 95% confidence interval. The maps represent the uncertainty in surface concentrations propagated from the uncertainties reported in the UK emissions (Wakeling et al., 2017) and estimated uncertainties in shipping emissions in the EMEP4UK model domain (Entec, 2002; Moreno-Gutiérrez et al., 2015). The uncertainties in surface concentration do not incorporate any uncertainties in the spatial and temporal aspects of the input emissions because no data on these aspects of uncertainty are provided by the
compilers of the emissions inventories.

For $O_3$ and $NO_2$ the areas with the highest uncertainty coincide with the location of the shipping lanes. This is due to assigning all shipping emissions an uncertainty of ± 30%, which causes high variability in the corresponding $NO_x$ emissions. The uncertainty in $O_3$ surface concentrations for the land-based grid cells is generally low (median relative uncertainty is ± 0.6%) with values of uncertainty up to ± 7% or ± 1.4 ppb occurring in the grid cells containing major UK cities. The overall low
uncertainty in the modelled $O_3$ concentrations can be attributed to the combination of a low uncertainty in precursor emissions and the substantial contribution of hemispheric background $O_3$ to UK ambient concentrations, the concentrations of which are not part of this analysis of uncertainty with respect to the UK-only emissions.

The relative uncertainty of $NO_2$ has a homogeneous spatial pattern (median relative uncertainty for all land-based grid cells is ± 7.4%) while absolute uncertainty is found to be higher (up to ± 3.5 μg m$^{-3}$ or ± 9%) in the areas with the major UK cities.
The magnitude of uncertainty in $NO_2$ is determined by the combination of two factors: i) $NO_2$ uncertainty is driven by $NO_x$ emission inputs which have low levels of uncertainty associated with them; ii) low overall variation in $O_3$ surface concentrations affects the reactions between NO, $NO_2$ and $O_3$ that are linked through the photolysis of $NO_2$ to give NO and the reaction of NO with $O_3$ to produce $NO_2$.

The spatial pattern of $PM_{2.5}$ surface concentrations and the corresponding absolute and relative uncertainties differ from those
for $O_3$ and $NO_2$. The concentration gradient indicates the presence of transboundary $PM_{2.5}$ transport into the UK. This is consistent with findings reported by AQEG (2013) that only about half of the $PM_{2.5}$ annual average concentrations have a UK origin. The spatial pattern of uncertainty in $PM_{2.5}$ concentrations shows higher uncertainty, both relative and absolute, in the grid cells with major cities; median relative uncertainty for all land-based grid cells is ± 4.6% with up to ± 9% (± 0.9 μg m$^{-3}$) in the grid cells with major cities. The surface concentrations of $PM_{2.5}$ are dominantly comprised of primary $PM_{2.5}$ emissions
and inorganic aerosols resulting from chemical reactions between $SO_2$, $NO_x$, and $NH_3$. Hence the spatial pattern of uncertainty can be explained by the fact that the main contribution to primary $PM_{2.5}$ comes from emissions from sources such as stationary combustion (e.g. residential heating) and road transport. The pattern of decreasing uncertainty from the land-based grid cells



(centre) towards the edges of the domain indicates the change in variation due to the transport of $PM_{2.5}$ away from the sources of emitted pollutants.

The overall uncertainty in the output variables ($O_3$, $NO_2$, and $PM_{2.5}$) was found to be lower compared to the uncertainty of the model input emissions. This can be explained by the overall weak response of surface concentrations to changes in the emission

originating from the UK which leads to the conclusion that the surface concentrations are affected by the transport of pollutants from elsewhere. Another explanation is the 'compensation of errors' whereby a positive effect of one or multiple input variables on the output is compensated by a negative effect of another input variable(s). This leads to the narrower confidence intervals associated with the EMEP4UK outputs.

An important observation from this uncertainty analysis is that the areas with the highest uncertainty coincide with the most

populated areas. Given that $O_3$, $NO_2$, and $PM_{2.5}$ are associated with adverse health effects, it is particularly important to have an estimate for the confidence level of the modelled values in the more densely-populated regions. This work has shown that the highest uncertainty is precisely in these regions. The reason for the increased levels of uncertainty in the grid cells coinciding with urban areas is discussed below.

## 3.2 Sensitivity analysis

In addition to quantitative uncertainty estimates, it is of interest to know how the uncertainty of each input contributes to the overall uncertainty and whether there are interactions between inputs that potentially affect the magnitude of overall uncertainty. This was achieved by conducting a variance-based sensitivity analysis.

Figures 3, 4, and 5 show the spatial distribution of the first-order sensitivity indices that represent the fractional contribution of the uncertainty of each input variable to the overall uncertainty in the output. Only the variables with $S_i > 0.03$ are presented

here. First-order indices with values less than 0.03 were omitted as the method used for computation of sensitivity indices is prone to numerical errors when the analytical sensitivity index values are close to zero (Saltelli et al., 2006). The threshold was estimated by examining the noise in first-order sensitivity indices calculated for unimportant input variables. Excluding $S_i < 0.03$ does not have an effect on the results presented because a relative contribution of less than 3% to the overall uncertainty can be considered negligible.

Difference between total and first-order sensitivity is used to highlight interactions between variable $X_i$ and all other input variables. For the sensitivity coefficients computed for the annual-averaged model outputs, there was no substantial difference found between first and total- order sensitivity indices, hence no between-input interactions were identified on the annual timescale (Fig. S2).

Figure 3 shows the spatial distribution of first-order sensitivity indices for the input variables affecting modelled $O_3$

concentrations. It is predominantly the $NO_x$ input emissions that drive the uncertainty in modelled $O_3$ surface concentrations. The greatest contribution to $O_3$ surface concentration uncertainty in the areas with higher levels of overall uncertainty is from the input variable $NO_x$_O, which represents $NO_x$ emissions from all the other SNAP sectors apart from SNAP 1 (combustion in energy and transformation industries) and SNAP 7 (road transport). The $NO_x$ emissions combined into this input variable





account for 27% of total $NO_x$ emissions and the uncertainty range for this variable is $\pm 19\%$. The input variable $NO_x\_D1$ (emissions from combustion in energy and transformation industries) does not contribute substantially to output uncertainty despite making up 41% of total $NO_x$ emissions, with a relative uncertainty of $\pm 7\%$. This is explained by the height at which these emissions occur; the emissions are injected into the vertical layers at heights of >184 m above ground level. This leads to $NO_x$ being dispersed and transported away from these elevated sources without affecting ground-level $O_3$ concentrations locally. The $NO_x$ emissions from input variable $NO_x\_D2$ (road transport) account for the remaining 32% of total $NO_x$ emissions. The spatial distribution of corresponding sensitivity indices indicates that uncertainty in road transport emissions affects overall uncertainty in $O_3$ surface concentrations in the grid cells closest to the emission sources (i.e. major roads). A large proportion (>80%) of overall uncertainty in $O_3$ concentrations in areas adjacent to the south and south-east coasts of England is apportioned to the uncertainty in shipping emissions.

In Scotland, most of the overall uncertainty in $O_3$ surface concentration is apportioned to the variables VOC_D and VOC_O that respectively represent VOC input emissions from the dominant VOC source sector (solvent and other product use) and emissions from the rest of the source sectors grouped into a single input. A small proportion is apportioned to the variable $NH3\_D$ that represents $NH_3$ emissions from agricultural sources. The effect of these input variables manifests in Scotland because of low levels of locally-emitted $NO_x$. The overall uncertainty in this area is very low.

In summary, the uncertainty in modelled surface concentrations of $O_3$ in the densely populated areas can be apportioned to the uncertainty in NOx emissions from non-dominant sources and uncertainty in shipping emissions.

The uncertainty in surface concentration of $NO_2$ was found to be driven mostly by uncertainty in $NO_x$ emissions (variables $NO_x\_D1$, $NO_x\_D2$, $NO_x\_O$) and shipping emissions (Fig. 4). Similarly to $O_3$, $NO_2$ is most sensitive to $NO_x$ emissions combined from all SNAP sectors apart from SNAP 1 (combustion in energy and transformation industries) and SNAP 7 (road transport). There is almost no sensitivity to $NO_x$ emissions from SNAP 1, for the same reason given above that these are elevated emissions. The sensitivity to $NO_x$ emissions from SNAP 7 is most pronounced close to the source of emissions (i.e. major roads and cities).

The similarity in spatial distribution of sensitivity indices for $O_3$ and $NO_2$ model outputs results from the concentrations of these pollutants being inversely correlated, as their chemical transformation reactions are interlinked. In the same way as for $O_3$, uncertainty in the $NO_2$ concentrations along the south and south-east coasts of England is mostly driven by the uncertainty in the shipping emissions. In fact, uncertainty in shipping emissions contributes approximately 30% of uncertainty in $NO_2$ concentrations even well inland, in areas away from major roads and cities.

Figure 5 shows the spatial distribution of first-order sensitivity indexes for the model inputs that contribute to the uncertainty in modelled surface concentrations of $PM_{2.5}$. Modelled $PM_{2.5}$ is sensitive to all emissions of $NH_3$ (dominant sector is agriculture) and to primary $PM_{2.5}$ (dominant sectors D1 is residential combustion and D2 is road transport), and to shipping emissions. In the areas with lower surface $PM_{2.5}$ concentrations such as Scotland, Wales, northern England and south-west England the uncertainty is mainly driven by $NH_3$ emissions from agriculture ($NH_3\_D$). The spatial pattern of emissions sensitivity indices for $PM_{2.5}$ mirrors the spatial distribution of $PM_{2.5}$ emission sources. From Figure 2 and Figure 5 it can be





seen that in the areas with the highest levels of uncertainty the model output is most sensitive to the emissions of primary $PM_{2.5}$. Similar to the results for $O_3$ and $NO_2$, the areas with the highest uncertainty coincide with the most populated areas.

The pattern in calculated sensitivity indices partially agrees with a previous study of changes in $PM_{2.5}$ surface concentrations in response to 30% reduction in emissions of $PM_{2.5}$, $NH_3$, $SO_x$, $NO_x$, and VOC by Vieno et al. (2016). In the study by Vieno et al. (2016) surface concentrations of $PM_{2.5}$ were found to be sensitive to reductions in each of the five pollutants individually (the same reduction was applied to a pollutant's emissions from all SNAP sectors simultaneously), with highest sensitivity to $NH_3$ and $PM_{2.5}$ emissions (up to ~6% reduction in surface concentration in response to 30% reduction in emissions). In comparison, our study the uncertainty in $PM_{2.5}$ surface concentrations is not affected by the perturbations of $SO_x$, $NO_x$, and VOC. This is likely to be due to i) the difference in ranges of variation (i.e. uncertainty ranges) in this study ($SO_x$, $NO_x$ and VOC input variables have narrower ranges of variation compared to $PM_{2.5}$ and $NH_3$), and ii) the presence of non-additivity and non-linearity in the model response to perturbations in the inputs.

### 3.3 Uncertainty propagation and sensitivity analysis for monthly averaged model outputs

The uncertainty assessment and sensitivity analysis for monthly averaged surface concentrations of $NO_2$, $O_3$, and $PM_{2.5}$ were performed for five different grid cells that were assigned the following environment types based on the national-network monitoring site within that grid cell: Auchencorth Moss and Harwell - rural background, Birmingham Acocks Green and London N. Kensington - urban background, and London Marylebone Road - urban traffic.

Monthly average concentrations with error bars representing the absolute uncertainty values (as a 95% CI) are presented in Figure 6. Figure 7 shows corresponding values of the relative uncertainty. Figure 8 shows how the magnitude of first-order sensitivity indices estimated for five different grid cells changes on a monthly timescale. If all first-order sensitivity coefficients add up to 1 then there are no interactions between inputs and all model variance can be apportioned to the variance in the individual inputs.

The $NO_2$ surface concentrations show a seasonal trend of lower concentrations occurring during summer months with the exception of the Auchencorth Moss grid cell where $NO_2$ concentrations are low throughout the year. The magnitude of uncertainty in $NO_2$ is proportional to the modelled concentration and changes relative to the concentration, which can be seen from the monthly relative uncertainty values (Fig. 7). The first-order sensitivity indices for $NO_2$ show that only $NO_x$ emissions (across all sectors) and shipping emissions influence the modelled surface $NO_2$ concentrations. Hence it can be concluded that the uncertainty in modelled concentrations of $NO_2$ directly depends on the uncertainty in $NO_x$ emissions and is not affected by the uncertainties in the emissions of any other pollutant. The change in the magnitude of sensitivity coefficients for the Harwell grid cell indicates increasing influence of shipping emissions on $NO_2$ concentrations during the summer months. Potential explanation for this is seasonal change in the wind direction which results in more $NO_x$ from shipping emissions being transported to the grid cell during the summer months.

The uncertainties in the $O_3$ modelled surface concentrations show an inverse seasonal trend compared to the uncertainties in modelled $NO_2$. Unlike the uncertainty in $NO_2$ concentration, the uncertainty in $O_3$ concentration is influenced by the grid cell





environment type; the highest level of uncertainty is observed for the London Marylebone Road grid cell (urban traffic). The relative uncertainty in $O_3$ concentrations for the Auchencorth Moss grid cell (rural background) is small and close to the median relative uncertainty in $O_3$ for annual average concentrations, which as discussed above is $\pm$ 0.6%. This indicates that perturbations in the input emissions do not substantially affect $O_3$ concentration in this grid cell. Although the magnitude of

uncertainty in $O_3$ is very small in this grid cell, the inputs that drive it differ noticeably throughout the year; during May-August the variance is mostly explained by VOC emissions (explains 77% of uncertainty for July) and during November-February $NO_x$ emissions drive the uncertainty. The magnitude of $O_3$ concentrations and corresponding uncertainties in the Birmingham Acocks Green and Harwell grid cells are very similar. The trends in sensitivity indices are also similar; during the April-September period some variance in the model output is explained by uncertainty in VOC emissions. However, in the

Harwell grid cell shipping emissions play a more important role. For the London-based grid cells, the level of uncertainty is the highest and it is mainly driven by the uncertainty in $NO_x$ and shipping emissions.

For the $PM_{2.5}$ monthly average concentrations, London-based grid cells show the highest values of absolute uncertainty and Auchencorth Moss - the lowest. The relative uncertainty in London based grid cells is also the highest. From Figure 7 it can be seen that the contribution to the overall uncertainty from the uncertainty due to $NH_3$ emissions for these grid cells is not as

important as for other three, the majority of uncertainty is explained by the uncertainty in the primary $PM_{2.5}$ emissions with $PM_{2.5}$ from road transport being the dominating variable. In Birmingham Acocks Green and Harwell, the effect of $NH_3$ emissions from agricultural sources is more pronounced; from 30% to 70% of overall uncertainty in $PM_{2.5}$ can be apportioned to uncertainty coming from agricultural emissions of $NH_3$ during spring and summer months.

### 3.4 Wider implications of our study

There are published studies that apply global sampling-based uncertainty and sensitivity analyses as well as derivative based methods (methods that do not have limitations of local OAT, i.e. linearity assumption) to ACTMs. However, the results reported by these studies are mostly of use for model development and calibration purposes and not the assessment of confidence in the model predictions/outputs. This is mainly because the simulations are performed for a short period ranging from days (Beddows et al., 2017; Chen and Brune, 2012; Rodriguez et al., 2007) to weeks (Cohan et al., 2010; Shrivastava et

al., 2016).

Additionally, in some studies, commercial software or packages with a graphical user interface (GUI) are used for global sensitivity and uncertainty analysis (Chen and Brune, 2012; Christian et al., 2017; Lee et al., 2011). These tools are well designed for a specific purpose but lack the option to scale up and to automate the analysis, i.e. ability to calculate sensitivity indices and uncertainty ranges for thousands of grid squares automatically.

Our study addresses both of the shortcomings. We demonstrate sensitivity and uncertainty analyses for the ACTM for a whole year for the UK domain as well as investigate variations in sensitivity and uncertainty on the monthly timescale for multiple locations with different environmental characteristics. Additionally, the package used to create Gaussian process emulators and to conduct uncertainty and sensitivity calculations is fully customisable and can be adapted for any application.



The model runs generated for the global sensitivity and uncertainty analysis can be utilised for other purposes provided that the sampling range for all inputs of interest is wide enough. For example, in our study the training points for the Gaussian emulator were selected to cover a wider range of input perturbations compared to the corresponding uncertainty range (Table 2). For all input emissions of $SO_x$, $NO_x$, VOC, and $NH_3$ the ranges of variation for the LHS design were set to $\pm$ 40% of their

baseline value, for primary $PM_{2.5}$ emissions the range was set to $\pm$ 75% and for shipping emissions from $-$ 40% to $+$ 100%. Hence the emulators created in this study using the model runs within the aforementioned input space can be used to investigate other scenarios of the model response to input emission perturbations with no extra computational cost. Hence, alternative ranges and probability distributions can be assigned to the model inputs to estimate the resulting output uncertainty or the emulator can be used for various emission reduction scenario analyses.

**4 Conclusions**

In this study, we have conducted global sensitivity and uncertainty analyses for the EMEP4UK Eulerian atmospheric chemistry transport model to quantify the uncertainty in surface concentrations of $O_3$, $NO_2$, and $PM_{2.5}$ and to identify the input emission variables that contribute the most to the uncertainty in each of the outputs. The uncertainty for model outputs was estimated from the uncertainties assigned to the UK emissions of $SO_2$, $NO_x$, $NH_3$, VOC, and primary $PM_{2.5}$ and documented in the UK

National Atmospheric Emissions Inventory. The benefit of conducting global sensitivity analysis in addition to uncertainty assessment is that it allows to determine how a model responds to the input perturbations within the ranges set by the input uncertainty estimates and consequently to identify the inputs which cause the variation in the model outputs (i.e. drive the uncertainty). The median values of the overall uncertainty calculated for the UK land-based grid cells for annual average surface concentrations of $O_3$, $NO_2$, and $PM_{2.5}$ were found to be in the ranges of $\pm$0.6%, $\pm$7.4%, and $\pm$4.6% respectively. This

indicates that the variation in the input data (i.e. emissions) does not cause a substantial variation in the outputs. Our results indicate, that this can likely be explained by variations in the other model input parameters such as chemical reaction rates, deposition velocities or physical constant values which might cause more variation in the model outputs. Alternatively, surface concentrations of the modelled pollutants in the UK may be dominated by the precursor emissions and long-range transport from outside the UK and are therefore relatively insensitive to changes in the UK emissions.

As a consequence, our results can provide more clarity about the confidence in modelled surface concentrations of pollutants that affect human health, especially in densely-populated urban areas. The results of our analysis indicate that modelled surface concentrations of $O_3$, $NO_2$, and $PM_{2.5}$ have the highest level of uncertainty in the grid cells comprising dense urban areas. The uncertainties of $O_3$, $NO_2$, and $PM_{2.5}$ in these grid cells reach $\pm$ 7%, $\pm$ 9%, and $\pm$ 9% respectively.

In addition to obtaining a quantitative estimate of the overall uncertainty, the input emissions that have the greatest influence

on the uncertainty in the modelled outputs were identified by performing a global variance-based sensitivity analysis. It was found that in urban areas uncertainty in $PM_{2.5}$ concentrations are driven by the uncertainty in primary $PM_{2.5}$ emissions. In contrast, in more remote areas $NH_3$ emissions had a stronger influence. Emissions of $NO_x$ combined from non-dominant sectors



(i.e. all sectors excluding energy production and road transport) were found to contribute the most to the uncertainty in both $O_3$ and $NO_2$ surface concentrations. Along the south and east coasts of England the uncertainty in shipping emissions contributed the most to the overall uncertainty in $O_3$ and $NO_2$ concentrations.

The comparison between first and total-order sensitivity indices did not indicate substantial interactions between the input variables for the model response on the annual timescale.

In our study we also demonstrated how the degree of uncertainty changes throughout the year by calculating uncertainty ranges for monthly-averaged surface concentrations of $O_3$, $NO_2$, and $PM_{2.5}$ for five selected grid cells. The global sensitivity conducted for monthly-averaged values showed seasonal trends in the type of input emissions that drive uncertainty in the surface concentrations.

The ability to estimate uncertainty in the predictions produced by a model is vital, because even low levels of uncertainty could be important in areas where the model yields predictions of surface concentrations that are close to limit values. This can lead to instances of exceedance due to the binary nature of limit value exceedance calculations, i.e. concentration is either over or under the limit. The sensitivity analysis should be an integral part of the assessment process applied *ex-ante* for the implementation of policy interventions, as it is also important to know which of the inputs contribute to the uncertainty in model outputs the most.

This work has demonstrated a global sensitivity and uncertainty analyses application for a Eulerian ACTM. The emulator-based approach used here is applicable to any other complex model and any type of model inputs such as emissions, physical constants or chemical reaction rate constants. The results of the analyses provide useful insights into the level of confidence in modelled predictions. Additionally, the Gaussian process emulators created for this analysis can be used with very little computational cost for any other scenario exploration purposes or assessment of overall uncertainty given different uncertainty ranges and probability distributions assigned to the model inputs.

*Data availability.* The EMEP MSC-W model code is available from www.github.com/metno/emep-ctm. The uncertainty and sensitivity data presented in this paper are available from https://doi.org/10.5281/zenodo.1321071 together with the analysis scripts.

*Competing interests.* The authors declare that they have no conflict of interest.

*Acknowledgements.* Ksenia Aleksankina acknowledges studentship funding from the University of Edinburgh and the NERC Centre for Ecology & Hydrology. The work by Stefan Reis and Massimo Vieno has been funded under the Natural Environment Research Council National Capability funding stream (Grant Reference ceh020011; Atmospheric Chemistry and Effects) and the Natural Environment Research Council Long-Term Science Single Centre funding stream UK-SCaPE.



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





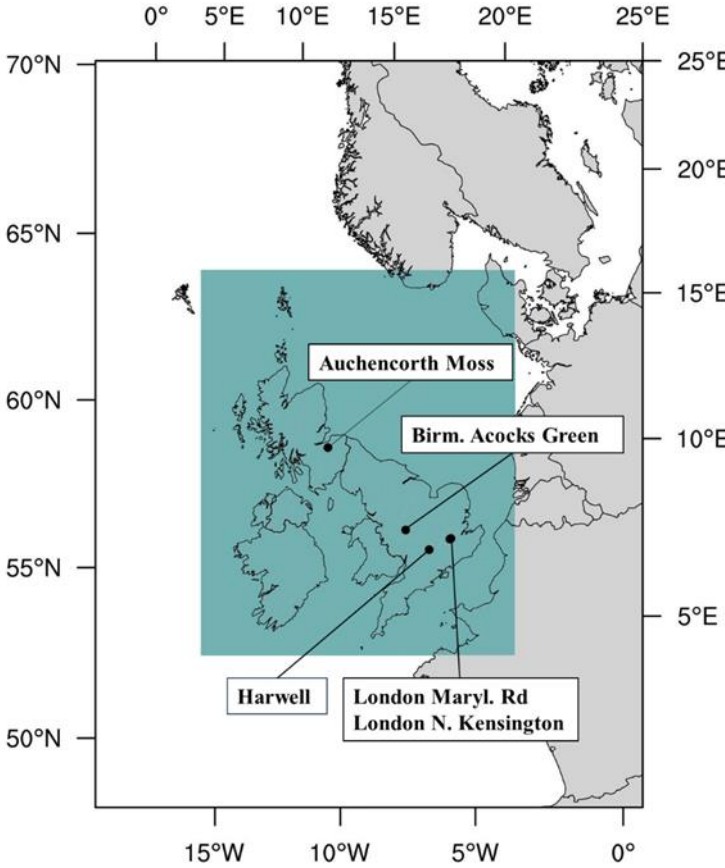

**Figure 1 The inner shaded box illustrates the EMEP4UK model British Isles domain, which is modelled at 5 km× 5 km horizontal resolution. The location of five grid cells used for uncertainty quantification and sensitivity analysis for monthly average modelled concentrations of O₃, NO₂, and PM₂.₅ are shown.**





**Figure 2 Baseline surface concentrations of O₃, NO₂, and PM₂.₅, and their respective spatial distributions of the absolute and relative uncertainties (at the 5 km × 5 km model grid resolution, year 2012) for the specified uncertainties in UK emissions. The uncertainty values are represented as a range of ± the baseline value and represent the 95% confidence interval.**





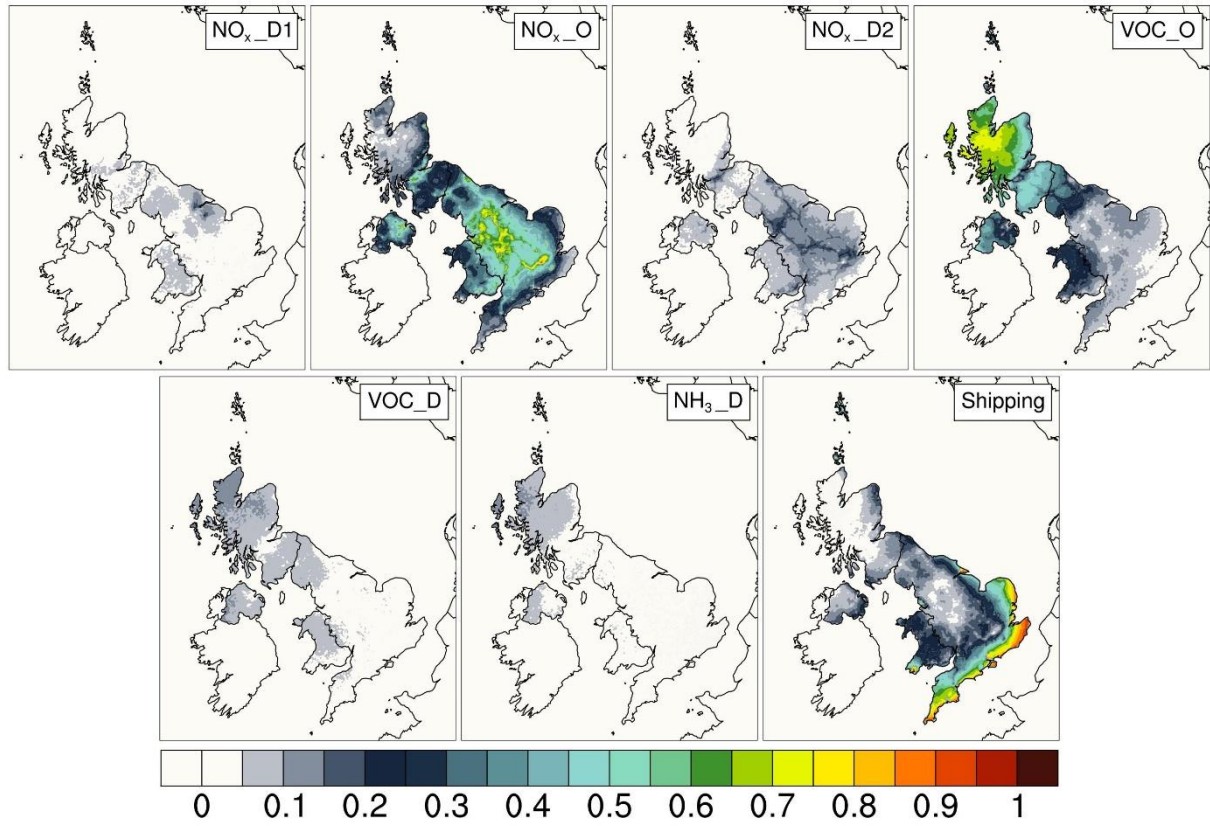

**Figure 3 Spatial distributions (at the 5km×5km model grid resolution) of the first-order sensitivity indices for modelled surface concentrations of O₃. D indicates emissions from a dominant sector and O indicates grouped emissions from the rest of the sectors. For NOₓ emissions dominant sectors are energy production (D1) and road transport (D2), for VOC emissions – solvent use, and for NH₃ – agriculture. Shipping emissions variable combines emissions of all relevant pollutants.**





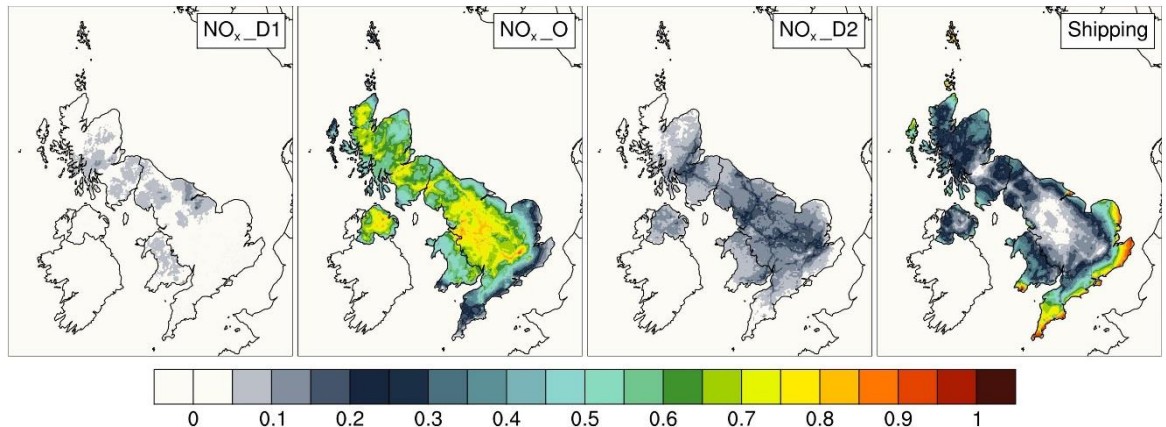

**Figure 4 Spatial distributions (at the 5km×5km model grid resolution) of the first-order sensitivity indices for modelled surface concentrations of NO₂. D indicates emissions from a dominant sector and O indicates grouped emissions from the rest of the sectors. For NO$_x$ emissions dominant sectors are energy production (D1) and road transport (D2). Shipping emissions variable combines emissions of all relevant pollutants.**



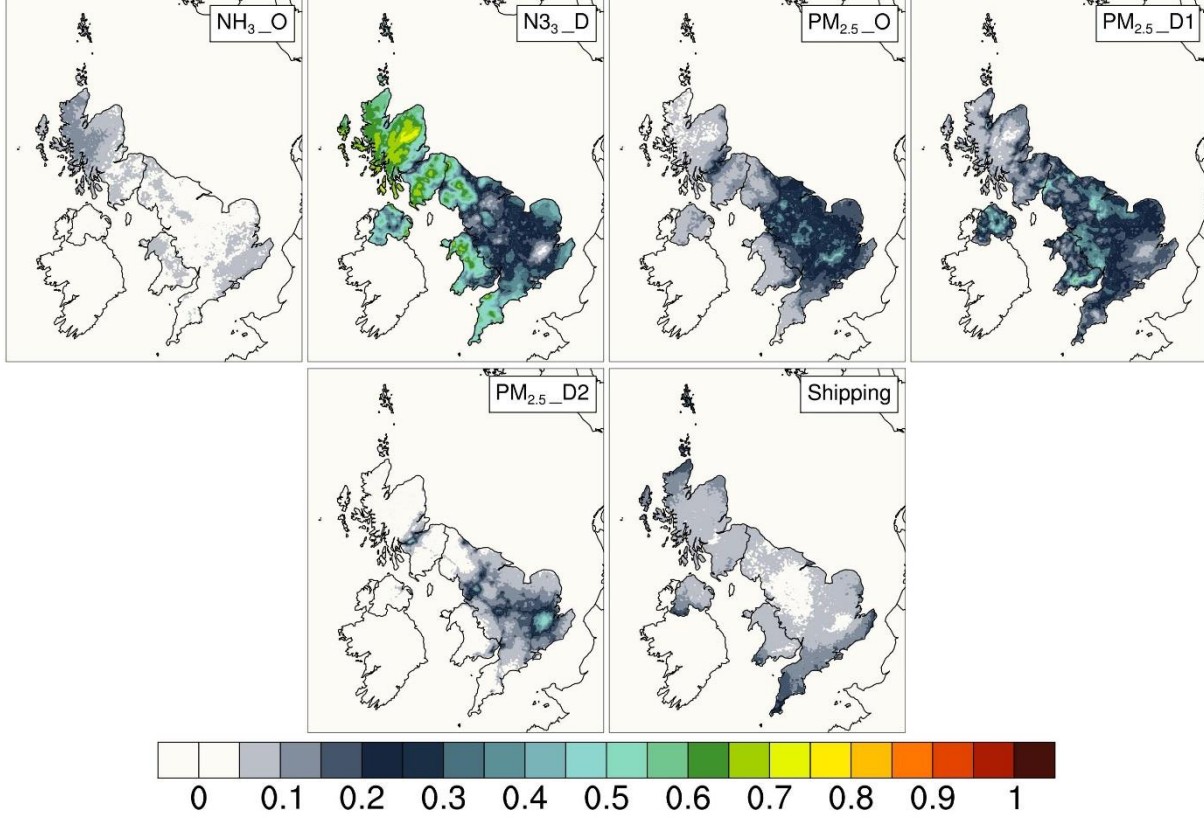

**Figure 5** Spatial distributions (at the 5km×5km model grid resolution) of the first-order sensitivity indices for modelled surface concentrations of PM$_{2.5}$. D indicates emissions from a dominant sector and O indicates grouped emissions from the rest of the sectors. For NH$_3$ emissions dominant sector is agriculture, for PM$_{2.5}$ dominant sectors are residential and non-industrial combustion (D1) and road transport (D2). Shipping emissions variable combines emissions of all relevant pollutants.





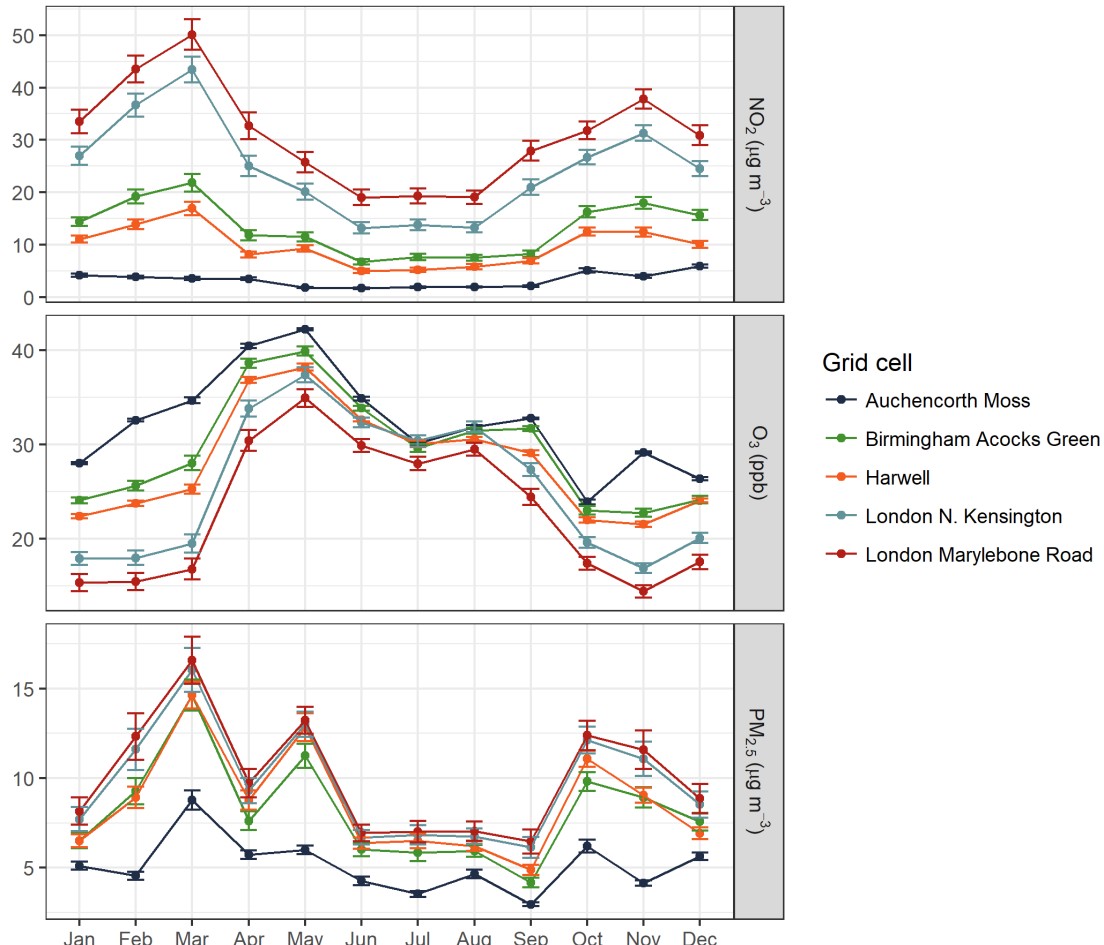

**Figure 6 Monthly average surface concentrations of NO₂, O₃ and PM₂.₅ with error bars showing (absolute) uncertainty, for five grid cells across the UK representing a spread of geographical locations and environment types. The environment types are assigned as follows: Auchencorth Moss and Harwell - rural background, Birmingham Acocks Green and London N. Kensington - urban background, and London Marylebone Road - urban traffic.**





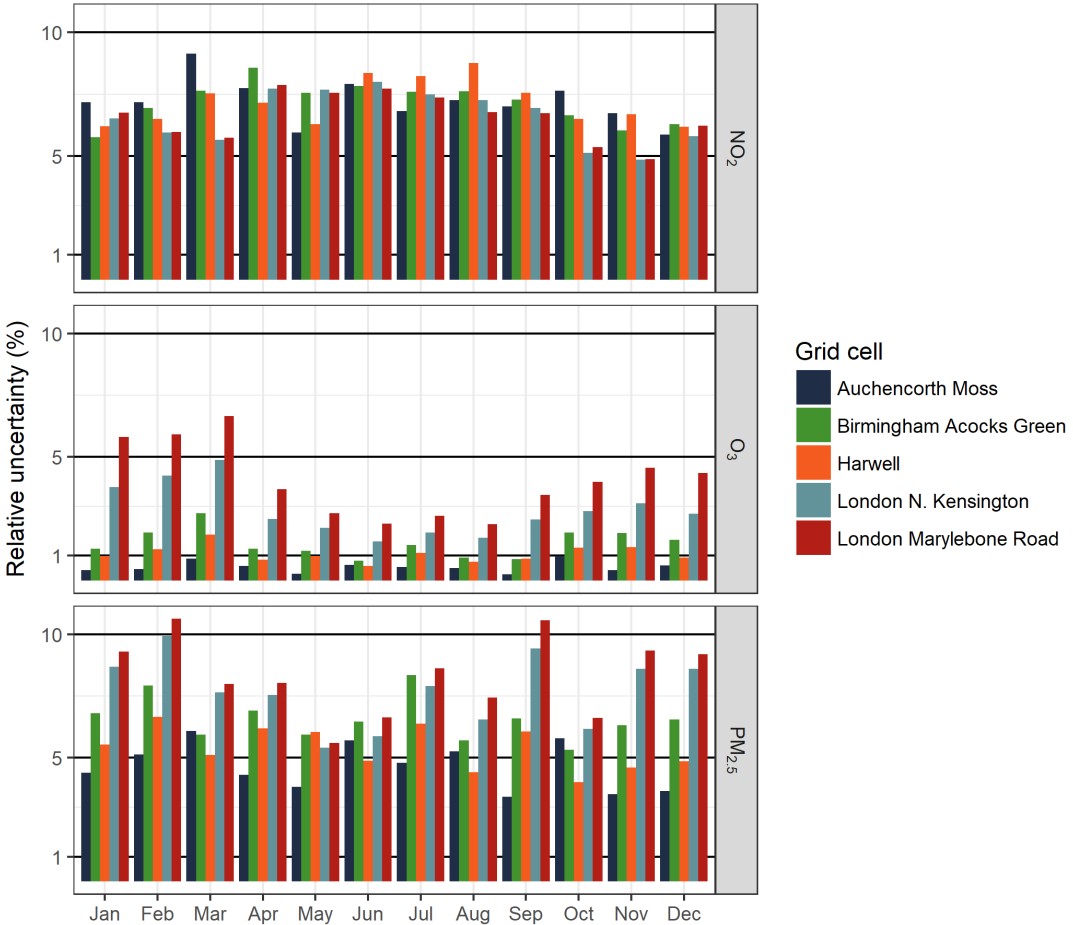

**Figure 7 Magnitude of relative uncertainty in monthly average surface concentrations of NO₂, O₃, and PM₂.₅ for five grid cells across the UK representing a spread of geographical locations and environment types. The environment types are assigned as follows: Auchencorth Moss and Harwell - rural background, Birmingham Acocks Green and London N. Kensington - urban background, and London Marylebone Road - urban traffic.**



**Figure 8 Monthly variation in the first-order sensitivity indices for five grid cells across the UK representing a spread of geographical locations and environment types. Based on the monitoring station classification grid squares are assigned the following environment types: Auchencorth Moss and Harwell - rural background, Birmingham Acocks Green and London N. Kensington - urban background, and London Marylebone Road - urban traffic.**



**Table 1 SNAP source sectors (Eurostat, 2004).**

| | |
|---|---|
| SNAP 1 | Combustion in energy and transformation industries |
| SNAP 2 | Residential and non-industrial combustion |
| SNAP 3 | Combustion in manufacturing industry |
| SNAP 4 | Production processes |
| SNAP 5 | Extraction and distribution of fossil fuels |
| SNAP 6 | Solvent and other product use |
| SNAP 7 | Road transport |
| SNAP 8 | Other mobile sources and machinery |
| SNAP 9 | Waste treatment and disposal |
| SNAP 10 | Agriculture |

**Table 2 Input variable definitions for the EMEP4UK uncertainty propagation and apportionment. The quoted uncertainties for emission sources are for UK annual totals. See main text for information on the sources of these values.**

| Variable used for sampling design | SNAP source sector | Contribution of source sector to total land-based emissions of that pollutant (%) | Uncertainty (as a 95% CI) | Ranges of scaling coefficients for the input emissions used in the LHS design |
|---|---|---|---|---|
| $SO_x\_D$ | 1 | 80 | ± 12 % | 0.6 – 1.4 |
| $SO_x\_O$ | 2-10 | 20 | ± 17 % | 0.6 – 1.4 |
| $NO_x\_D1$ | 1 | 41 | ± 7 % | 0.6 – 1.4 |
| $NO_x\_D2$ | 7 | 32 | ± 7 % | 0.6 – 1.4 |
| $NO_x\_O$ | 2-6, 8-10 | 27 | ± 19 % | 0.6 – 1.4 |
| VOC_D | 6 | 39 | ± 22 % | 0.6 – 1.4 |
| VOC_O | 1-5, 7-10 | 61 | ± 24 % | 0.6 – 1.4 |
| $NH_3\_D$ | 10 | 88 | ± 33 % | 0.6 – 1.4 |
| $NH_3\_O$ | 1-9, 10 | 12 | ± 35 % | 0.6 – 1.4 |
| $PM_{2.5}\_D1$ | 2 | 33 | ± 59 % | 0.25 – 1.75 |
| $PM_{2.5}\_D2$ | 7 | 21 | ± 59 % | 0.25 – 1.75 |
| $PM_{2.5}\_O$ | 1, 3-6, 8-10 | 46 | ± 58 % | 0.25 – 1.75 |
| Shipping | N/A | N/A | ± 30 % | 0.6 – 2.0 |

