# Peer review of "Advanced methods for uncertainty assessment and global sensitivity analysis of a Eulerian atmospheric chemistry transport model"

_Atmospheric Chemistry and Physics, 2018_

## Referee Comment (RC1) · Anonymous Referee #1 · 26 Aug 2018

General Comments

The authors have made a commendable effort to apply uncertainty and sensitivity analysis methods which have a long theoretical history in the stats literature but have only in recent years begun to be applied to complex models such as this.

Given that they only look at sensitivity to emissions, and conclude that the model is not particularly sensitive to those inputs, it is perhaps a shame that they didn't attempt to include more input variables into the analysis, as there have been a number of published studies which demonstrate that these methods can be used with significantly larger numbers of inputs.

[Figure]

There are a few concerns regarding the implementation of the methods and the effect that this may have on the validity of the results. In particular please see the points below concerning sample size and emulator validation, which should be addressed before the paper is recommended for publication.

Specific Comments

P5,L2 How many emissions inputs does the model have, and why were the ones used chosen?

P5,L5 Why were the shipping emissions not split by pollutant type when the other emissions were? Does this not make the results harder to interpret?

P6,L19 Could the authors include their reasoning for choosing only 84 design points. The normal recommended minimum number for constructing Gaussian process emulators is 10 per input variable, which in this case would be 130. See, for example Loeppky et al, 2009, Choosing the Sample Size of a Computer Experiment: A Practical Guide, Technometrics.

P6,L30 The authors state that the choice of a linear mean function incorporates "prior beliefs" – could they explain what prior beliefs motivated this choice.

P7,L10 The authors state that the emulator error was estimated using cross validation and this is presented in the SI. However very little detail is given there except a reference to a paper describing the Matlab package used to construct the emulators (Lataniotis, 2017), which says that the package uses cross validation for parameter estimation, not emulator validation. The authors also say in the paragraph above that they used cross validation for parameter estimation. A clear statement is required as to whether or not the same cross validation was used for both parameter estimation and emulator validation. The accuracy of the emulators is of such key importance to everything that follows that summary statistics of either a separate cross validation, or a validation with a held out data set, must be presented in the main text.
P8,L2 Could the authors comment on the validity of describing the 5km grid square containing Marylebone Rd as 'urban traffic' and the one containing N. Kensington as 'urban background'. Surely at this resolution both grid squares must be considered as urban background – this is demonstrated by the almost identical sensitivity of NO2 and O3 concentrations to NOx emissions from road transport shown in figure 8.

P8,L5 The authors state that the "sensitivity indices were estimated". There are a numbered of published methods for doing this so could they say which method they used and why.

P14,L9 Given that O3 concentration is known to be highly non-linear and non-monotonic in response to changes in NOx and VOC emissions, the contention that these emulators could be used to test emission reduction scenarios is questionable as the small number of training runs used to construct the emulators means they would be unlikely to be able to accurately predict the emissions levels at which the chemical regime changes between being NOx sensitive and VOC sensitive.

P14,L19 Could the very low sensitivity of annual mean ozone to emissions be a result of the strong diurnal variation and photochemical nature of the production of this pollutant? Could the authors comment on whether annual average 8-hour maximum might have been a more informative metric to emulate.

P14,L20 As the authors findings that variation in emissions does not cause substantial variation in the outputs contrasts with their statement in the introduction that it has been previously found that uncertainties in input emissions are major contributors to the uncertainty in the ACTM outputs, could they comment on why this might be – do they think it is a feature of the EMEP model or a result of the analysis methods?

P14,L25 Given the authors assertion in the previous paragraph that the uncertainty in model output is likely to be driven mainly by variables that they have not included in their analysis, would they concede that their uncertainty estimates are likely to under-estimate by a large degree the real uncertainty in the model output, i.e. that caused

buy uncertainty in all of the input variables plus the model discrepancy.

---

## Referee Comment (RC2) · Anonymous Referee #2 · 24 Oct 2018

Review of "Advanced methods for uncertainty assessment and global sensitivity analysis of a Eulerian atmospheric chemistry transport model" by Ksenia Aleksankina, Stefan Reis, Massimo Vieno, and Mathew R. Heal

This discussion paper by Aleksankina et al. documents a global sensitivity and uncertainty analyses for the regional chemical transport model EMEP4UK, with the objective of quantifying the uncertainty in surface concentrations of air pollutants (ozone, nitrogen dioxide, and particulate matter below 2.5 um in diameter) and the contribution to that uncertainty from uncertainties in UK-only emissions. No uncertainties associated with model transport and/or chemical processes, or the lateral boundary conditions or

driving meteorology were considered.

I found the paper to be well organised, well written, and a really nice example of applying powerful statistical approaches to understanding model behaviour and uncertainties. The discussion on the sensitivity analysis itself was very interesting and shows how insightful this technique is. The paper will add to the growing literature base on the use of Gaussian emulation in quantifying uncertainties in geophysical models. I wholeheartedly recommend that the paper is accepted and published in Atmos. Chem. Phys. However, I have a few comments which I hope the authors will consider when submitting a revised manuscript:

1. Intro: For the non-specialist, I think it would be worthwhile to include some basic introductory material on what you mean by sensitivity analysis versus uncertainty analysis.

2. Can you include some discussion on structural uncertainty?

3. Intro: Note that aerosols affect climate through aerosol-cloud interactions and not only aerosol-radiation interactions

4. Intro: Meta models have also been used in exploring climate sensitivity/climate response e.g. Murphy et al. (2004)

5. Section 2.1: Full names for SO2, NH3 etc..

6. Section 2.1: Can you include details of bvoc emissions scheme, and parameterisations for sea salt and dust emissions?

7. Table 2: Slight error with SNAP sectors for NH3_O (i.e. 10 should not be included!)

8. Results Section 3.1: You say that there is a "substantial contribution of hemispheric background O3 to UK ambient concentrations"? Can you be more quantitative here?

9. Results Section 3.1: You refer to the 'compensation of errors' as one explanation why the surface response is weak given the input uncertainties. Can you point to the

literature for evidence of this statement? I've only seen "compensation of errors" only referred to in the context of process representation in models.

10. Results Section 3.3: One potential explanation for the seasonal change in sensitivity at Harwell to shipping emissions is the seasonal change in the wind direction which results in more NOx from shipping emissions being transported to the site. Can this be verified from the WRF meteorology used to drive the model?
* * *

---

## Author Comment (AC1) · 13 Dec 2018

**acp-2018-690: Advanced methods for uncertainty assessment and global sensitivity analysis of a Eulerian atmospheric chemistry transport model**

by Aleksankina et al.

**Response to reviewer #1**

*General Comments*

*The authors have made a commendable effort to apply uncertainty and sensitivity analysis methods which have a long theoretical history in the stats literature but have only in recent years begun to be applied to complex models such as this.*

**Response:** We thank the reviewer for this supportive comment.

1) *Given that they only look at sensitivity to emissions, and conclude that the model is not particularly sensitive to those inputs, it is perhaps a shame that they didn't attempt to include more input variables into the analysis, as there have been a number of published studies which demonstrate that these methods can be used with significantly larger numbers of inputs.*

**Response:** There is still a practical trade-off between the desirability of investigating a large number of inputs and the computational costs associated with running a complex atmospheric chemistry transport model (ACTM) for that number of inputs. Hence in this study we concentrated on the input emissions, because input emissions have been reported to strongly affect uncertainty in the modelled surface concentrations of various air pollutants and because simulation of the effects of changes in emissions is a principal application for an ACTM (Introduction, p2 L17). It is only by deploying formal sensitivity methods as we have done here that robust statements can ultimately be made concerning the sensitivities of a given ACTM – in our illustrative case here, the sensitivities of the EMEP4UK model to input emissions.

2) *There are a few concerns regarding the implementation of the methods and the effect that this may have on the validity of the results. In particular please see the points below concerning sample size and emulator validation, which should be addressed before the paper is recommended for publication.*

**Response:** We address these points where they are raised in more detail below.

*Specific Comments*

3) *P5,L2 How many emissions inputs does the model have, and why were the ones used chosen?*

**Response:** The model uses both anthropogenic and biogenic emissions as inputs – a full description of the model inputs can be found in Simpson et al. (2012). In this study, emissions of all of the major primary anthropogenic pollutant compounds were investigated,

as described in the paper. The biogenic emission sources were not perturbed, as these are computed within the model itself and are not therefore classed as input datasets. This decision was made based on the fact that one of the main applications of the EMEP model is in providing scientific support for policy-making regarding impacts of interventions leading to anthropogenic emissions reductions (see Introduction). In the short to medium term at least, the potential future changes in emissions driven by environmental and climate change policies are not likely to affect biogenic emissions as much as anthropogenic emissions, hence it was decided to investigate model response to the changes in anthropogenic emissions.

4) *P5,L5 Why were the shipping emissions not split by pollutant type when the other emissions were? Does this not make the results harder to interpret?*

**Response:** This goes back to the trade-off between how many input variables should be used in the analysis to obtain the most benefit from it and the computational cost of the analysis. The decision was made to concentrate on the land-based input emissions. The interpretation of sensitivities to shipping emissions was made harder by our approach only in places particularly impacted by shipping. The majority of this impact takes place in the sea-based grid cells where the assessment of the impact of air pollutants on human health and ecosystems is not relevant.

5) *P6,L19 Could the authors include their reasoning for choosing only 84 design points. The normal recommended minimum number for constructing Gaussian process emulators is 10 per input variable, which in this case would be 130. See, for example Loeppky et al, 2009, Choosing the Sample Size of a Computer Experiment: A Practical Guide, Technometrics.*

**Response:** We were aware of this paper cited by the reviewer. The paper states that an empirical $n = 10d$ rule (where $n$ is the sample size and $d$ is the number of variables under investigation) provides reasonable accuracy when approximating model response with a Gaussian process emulator. This rule is applicable to the model response of any complexity. The paper indicates this is an empirical rule and does not state that $10d$ is the minimum sample size for a Gaussian process emulator to perform well. In our particular study, the input-output response function is expected to be smooth, hence there is no need for a very large number of sample points as it will not reduce error of the emulator.

Again, it is a balance on return for computational resource. Ideally, in the situation where model runs are computationally expensive a sequential sampling technique can be applied to track the improvement of emulator performance with increase in the sample size.

6) *P6,L30 The authors state that the choice of a linear mean function incorporates "prior beliefs" – could they explain what prior beliefs motivated this choice.*

**Response:** We are referring here to the fact that whilst one can choose the form of $h(.)$ to use, the choice incorporates assumptions, i.e. prior beliefs, as to the most appropriate form to use. In this study the choice was a linear mean function. We assumed that the response of the

surface concentration of air pollutants to the changes in the input emissions is likely to be smooth (i.e. with no discontinuities and no sharp fluctuations). Therefore, a prior linear trend was chosen to be more suitable compared to using a constant (mean) or a quadratic (or higher polynomial) function as a trend. In the revised paper we have now expanded the relevant sentence as follow (p7, L12) to explain the choice:

"In this study, the mean function was chosen to have a linear form $\beta_o + \sum_{i=1}^{13} \beta_i x_i$, on the basis that the response of the surface concentration to changes in input emissions is expected to be smooth with no discontinuities or fluctuations."

7) *P7,L10 The authors state that the emulator error was estimated using cross validation and this is presented in the SI. However very little detail is given there except a reference to a paper describing the Matlab package used to construct the emulators (Lataniotis, 2017), which says that the package uses cross validation for parameter estimation, not emulator validation. The authors also say in the paragraph above that they used cross validation for parameter estimation. A clear statement is required as to whether or not the same cross validation was used for both parameter estimation and emulator validation. The accuracy of the emulators is of such key importance to everything that follows that summary statistics of either a separate cross validation, or a validation with a held-out data set, must be presented in the main text.*

**Response:** We have checked the cross-validation error values by performing a separate cross-validation calculation. The results of explicitly-performed cross-validation do not differ from that which was originally presented in the supplementary material. We accept that the reference we originally cited may have caused some confusion, as the formula for cross-validation is reported in the parameter estimation section. The same formula was used for calculating the cross-validation error for the emulator. We now instead explicitly state in the supplementary information the cross-validation equation used. The three figures in the supplementary information have now been replaced with the updated cross-validation versions. The code written to perform these independent cross-validations is also added to the data repository for this paper.

8) *P8,L2 Could the authors comment on the validity of describing the 5km grid square containing Marylebone Rd as 'urban traffic' and the one containing N. Kensington as 'urban background'. Surely at this resolution both grid squares must be considered as urban background – this is demonstrated by the almost identical sensitivity of $NO_2$ and $O_3$ concentrations to $NO_x$ emissions from road transport shown in figure 8.*

**Response:** The descriptors assigned to the grid squares are the descriptors used by Defra and the UK air quality community for the national network air quality monitoring stations in those grid squares. Whilst we agree that the 5 km resolution of the model causes some averaging of modelled pollutant output compared with a point measurement at those monitoring stations, we do not agree that our results are almost identical for the two grids. The modelled concentrations of $NO_2$ and $O_3$ as well as the uncertainty values are different for the two grid squares, and the sensitivity indices (fig. 8) are similar but not identical. It is not

claimed that the "Marylebone Rd" patterns of sensitivity are applicable to all 'urban traffic' sites.

9) *P8,L5 The authors state that the "sensitivity indices were estimated". There are a numbered of published methods for doing this so could they say which method they used and why.*

**Response:** The references provided in the paper describe the numerical methods used for the estimation of the first and total order indices. These methods for estimation of the sensitivity indices were chosen as they are widely used and well-established methods. However, to clarify explicitly the methods we used for these indices we have now added the following sentence to section 2.4.2: "The first and total-order sensitivity indices were estimated following the methods described by Sobol' (1993) and Janon et al. (2014) respectively."

10) *P14,L9 Given that $O_3$ concentration is known to be highly non-linear and nonmonotonic in response to changes in NOx and VOC emissions, the contention that these emulators could be used to test emission reduction scenarios is questionable as the small number of training runs used to construct the emulators means they would be unlikely to be able to accurately predict the emissions levels at which the chemical regime changes between being NOx sensitive and VOC sensitive.*

**Response:** The response of $O_3$ to changes in $NO_x$ and VOC concentrations although non-linear is still smooth. The effect on $O_3$ is analysed and presented on annual and monthly timescales, hence additional hourly and daily fluctuations are smoothed out.

11) *P14,L19 Could the very low sensitivity of annual mean ozone to emissions be a result of the strong diurnal variation and photochemical nature of the production of this pollutant? Could the authors comment on whether annual average 8-hour maximum might have been a more informative metric to emulate.*

**Response:** The annual average 8-hour metric and the annual average are very strongly correlated metrics (see, for example, page 34 of AQEG (2009)); so whilst the absolute values will change we do not believe the nature of the $O_3$ response to the input emission perturbations will change.

12) *P14,L20 As the authors findings that variation in emissions does not cause substantial variation in the outputs contrasts with their statement in the introduction that it has been previously found that uncertainties in input emissions are major contributors to the uncertainty in the ACTM outputs, could they comment on why this might be – do they think it is a feature of the EMEP model or a result of the analysis methods?*

**Response:** As was noted in the paper, the previous studies we refer to were performed with different models. The lack of sensitivity here could be due both to the fact that annual and monthly averaged surface concentrations were investigated and to the possible strong effect

of background concentrations of pollutants as well as pollutants transported from outside the UK.

The point of this study was not to show how good or not the EMEP4UK model is at reproducing observed pollutant concentrations but to demonstrate a method that can be used to improve the understanding of model response and/or to potentially point out the aspects that may need attention or improvement in future model development.

13) *P14,L25 Given the authors assertion in the previous paragraph that the uncertainty in model output is likely to be driven mainly by variables that they have not included in their analysis, would they concede that their uncertainty estimates are likely to underestimate by a large degree the real uncertainty in the model output, i.e. that caused by uncertainty in all of the input variables plus the model discrepancy.*

**Response:** Yes, we agree that the total uncertainty in the estimated concentration of the pollutants is higher than the uncertainty propagated from the input emissions only as there are other uncertain model inputs and parameters. Ideally sensitivity analysis should be incorporated as a part of the model development process (which could aid in both model simplification/reduction and calibration). In that approach the effect of all uncertain inputs and parameters could be assessed without having to do it retrospectively. In addition, screening techniques, e.g. the Morris method (Morris, 1991), could be applied to identify the inputs and parameters that most drive variation in the model outputs and which need to be investigated further. However, here we concentrate on presenting the application of the method itself and on a subset of inputs which previously was shown to drive uncertainty in the model output values.

**References:**

AQEG: Ozone in the United Kingdom. Fifth report of the Air Quality Expert Group., UK Department for Environment, Food and Rural Affairs, London. PB13216. ISBN 978-0-85521-184-4. http://uk-air.defra.gov.uk/library/reports?report_id=544, 2009.

Janon, A., Klein, T., Lagnoux, A., Nodet, M. and Prieur, C.: Asymptotic normality and efficiency of two Sobol index estimators, ESAIM Probab. Stat., 18, 342–364, doi:10.1051/ps/2013040, 2014.

Morris, M. D.: Factorial Sampling Plans for Preliminary Computational Experiments, Technometrics, 33(2), 161, doi:10.2307/1269043, 1991.

Simpson, D., Benedictow, a., Berge, H., Bergström, R., Emberson, L. D., Fagerli, H., Flechard, C. R., Hayman, G. D., Gauss, M., Jonson, J. E., Jenkin, M. E., Nyíri, a., Richter, C., Semeena, V. S., Tsyro, S., Tuovinen, J.-P., Valdebenito, Á. and Wind, P.: The EMEP MSC-W chemical transport model – technical description, Atmos. Chem. Phys., 12(16), 7825–7865, doi:10.5194/acp-12-7825-2012, 2012

Sobol', I. M.: Sensitivity estimates for nonlinear mathematical models, Math. Model. Comput. Exp., 1(4), 407–414, 1993.)

---

## Author Comment (AC2) · 13 Dec 2018

**acp-2018-690: Advanced methods for uncertainty assessment and global sensitivity analysis of a Eulerian atmospheric chemistry transport model**

by Aleksankina et al.

Response to reviewer #2

*This discussion paper by Aleksankina et al. documents a global sensitivity and uncertainty analyses for the regional chemical transport model EMEP4UK, with the objective of quantifying the uncertainty in surface concentrations of air pollutants (ozone, nitrogen dioxide, and particulate matter below 2.5 um in diameter) and the contribution to that uncertainty from uncertainties in UK-only emissions. No uncertainties associated with model transport and/or chemical processes, or the lateral boundary conditions or driving meteorology were considered. I found the paper to be well organised, well written, and a really nice example of applying powerful statistical approaches to understanding model behaviour and uncertainties. The discussion on the sensitivity analysis itself was very interesting and shows how insightful this technique is. The paper will add to the growing literature base on the use of Gaussian emulation in quantifying uncertainties in geophysical models. I wholeheartedly recommend that the paper is accepted and published in Atmos. Chem. Phys.*

**Response:** We much appreciate the reviewer's comments on the merits of the paper and their enthusiastic recommendation for its acceptance and publication. Thank you.

*However, I have a few comments which I hope the authors will consider when submitting a revised manuscript:*

1) *Intro: For the non-specialist, I think it would be worthwhile to include some basic introductory material on what you mean by sensitivity analysis versus uncertainty analysis.*

**Response:** The following text has been added to the Introduction (p2, L26):

"The main distinction between uncertainty and sensitivity analysis is that uncertainty analysis is performed to quantify model output uncertainty arising from the uncertainty in a single or multiple inputs, whilst sensitivity analysis is performed to investigate input–output relationships and to apportion the variation in model output to different inputs. Hence the sensitivity analysis allows conclusions to be drawn on the extent to which the overall variation in the modelled values is driven by variation in different inputs (Saltelli, 2002)"

*2. Can you include some discussion on structural uncertainty?*

**Response:** The following text has been added to the Introduction (p2 L17).

"There are various sources of uncertainty in a model; the sources range from structural or conceptual uncertainties about how well a given model represents reality to uncertainties in the model input data and physical and chemical constants, which have an effect on calculation results of the model."

3. *Intro: Note that aerosols affect climate through aerosol-cloud interactions and not only aerosol-radiation interactions*

**Response:** The first paragraph of the Introduction has been amended to now read as follows:

"Additionally, particulate matter and $O_3$ contribute to climate change through radiative forcing and aerosol-cloud interactions (for PM) (IPCC, 2013; Stevenson et al., 2013) and $O_3$ has an adverse impact on natural and semi-natural vegetation and crop yields (Teixeira et al., 2011)."

4. *Intro: Meta models have also been used in exploring climate sensitivity/climate response e.g. Murphy et al. (2004)*

**Response:** From our search of the literature we are assuming the reviewer is referring to the following paper: "Murphy, J. M., D. M. H. Sexton, D. N. Barnett, G. S. Jones, M. J. Webb, M. Collins, and D. A. Stainforth. Quantification of modelling uncertainties in a large ensemble of climate change simulations. Nature, 430, 768–772, 204." If so, we do not think it appropriate to include as an example of meta-model application in sensitivity analysis because the methodology for sensitivity analysis described in this paper is one-at-a-time (OAT) and is based on an ensemble approach.

5) *Section 2.1: Full names for SO2, NH3 etc.*

**Response:** Full names of chemical species are now added in the methods section where they first appear (p4, L19).

6. *Section 2.1: Can you include details of bvoc emissions scheme, and parameterisations for sea salt and dust emissions?*

**Response:** The following two blocks of text have now been added to section 2.1 (p4):

"Biogenic emissions of monoterpenes and isoprene are calculated by the model for every grid cell and time step according to the methodology of Guenther et al. (1993, 1995), using near-surface air temperature and photosynthetically active radiation as well as aggregated land-cover categorisations, as described in Simpson et al. (2012)."

And

"The details of the sea-salt generation parameterisation scheme used in the model are presented in Monahan et al. (1986) and Mårtensson et al. (2003). The boundary condition monthly average concentrations of fine and coarse dust are calculated with the global chemical transport model of the University of Oslo (Grini et al., 2005). The detailed parametrisation of dust mobilisation is presented in Simpson et al. (2012)."

7) *Table 2: Slight error with SNAP sectors for NH3_O (i.e. 10 should not be included!)*

**Response:** Thank you for spotting this typo which we have now corrected.

8) *Results Section 3.1: You say that there is a "substantial contribution of hemispheric background $O_3$ to UK ambient concentrations"? Can you be more quantitative here?*

**Response:** We cannot estimate the first-order effect of the background $O_3$ on the UK surface concentrations of $O_3$ as this input was not one of the perturbed inputs in this study. (We focused on perturbation of primary anthropogenic emissions.) It is not at all straightforward to quantify the background contribution to a secondary pollutant such as $O_3$ because not only is there import of $O_3$ and of $O_3$ precursors into the UK, but the UK is also a surface sink for $O_3$. The statement in our paper was based on Simpson et al. (2012) who state that "ambient ozone levels in Europe are typically not much greater than the Northern hemispheric background ozone". Additionally, in EMEP4UK a "Mace-Head" adjustment is applied to monthly boundary condition values of the $O_3$ concentrations. Hence in this paper the contribution of hemispheric background $O_3$ to the UK ambient concentrations is offered as a possible explanation of the lack of sensitivity of the surface $O_3$ to changes in the precursor emissions.

We have now added the citation to Simpson et al. (2012) to the end of the sentence in question.

9) *Results Section 3.1: You refer to the 'compensation of errors' as one explanation why the surface response is weak given the input uncertainties. Can you point to the literature for evidence of this statement? I've only seen "compensation of errors" only referred to in the context of process representation in models.*

**Response:** The phrase comes from Skeffington et al. 2007. In that paper the reason for narrowing of confidence limits for critical loads compared to those of the input parameters was explained to be due to a "compensation of errors" mechanism, but no further explanation was provided. Here, by compensation of errors we mean a situation when the variation in the output is less than expected. This could be caused by multiple inputs having an opposite effect on the magnitude of change in the output of interest.

We have changed the phrasing in our paper to "so-called compensation of errors".

10. *Results Section 3.3: One potential explanation for the seasonal change in sensitivity at Harwell to shipping emissions is the seasonal change in the wind direction which results in more NOx from shipping emissions being transported to the site. Can this be verified from the WRF meteorology used to drive the model?*

**Response:** The seasonal wind speed and direction for the year 2012 is shown in Figure A below, using the meteorology supplied from the AURN data as extracted using the openair package. It could be argued that there is some correlation between sensitivity index patterns in Figure 8 of our paper and the wind direction; however most likely the seasonality in $NO_x$ sensitivity to shipping emissions is due to a combination of interacting processes within the model.

[Figure]

FIGURE A. WIND ROSE, HARWELL AURN SITE, 2012.

**References:**

Grini, A., Myhre, G., Zender, C. S. and Isaksen, I. S. A.: Model simulations of dust sources and transport in the global atmosphere: Effects of soil erodibility and wind speed variability, J. Geophys. Res., 110(D2), D02205, doi:10.1029/2004JD005037, 2005.

Guenther, A., Hewitt, C. N., Erickson, D., Fall, R., Geron, C., Graedel, T., Harley, P., Klinger, L., Lerdau, M., Mckay, W. A., Pierce, T., Scholes, B., Steinbrecher, R., Tallamraju, R., Taylor, J. and Zimmerman, P.: A global model of natural volatile organic compound emissions, J. Geophys. Res., 100, 8873-8892, 1995.

Guenther, A. B., Zimmerman, P. R., Harley, P. C., Monson, R. K. and Fall, R.: Isoprene and Monoterpene Emission Rate Variability - Model Evaluations and Sensitivity Analyses, J. Geophys. Res., 98, 12609-12617, 1993.

Mårtensson, E. M., Nilsson, E. D., de Leeuw, G., Cohen, L. H. and Hansson, H.-C.: Laboratory simulations and parameterization of the primary marine aerosol production, J. Geophys. Res. Atmos., 108(D9), n/a-n/a, doi:10.1029/2002JD002263, 2003.

Monahan, E. C., Spiel, D. E. and Davidson, K. L.: A Model of Marine Aerosol Generation Via Whitecaps and Wave Disruption, pp. 167–174, Springer, Dordrecht., 1986.

Simpson, D., Benedictow, A., Berge, H., Bergström, R., Emberson, L. D., Fagerli, H., Flechard, C. R., Hayman, G. D., Gauss, M., Jonson, J. E., Jenkin, M. E., Nyiri, A., Richter, C., Semeena, V. S., Tsyro, S., Tuovinen, J. P., Valdebenito, A. and Wind, P.: The EMEP MSC-W chemical transport model - technical description, Atmos. Chem. Phys., 12, 7825-7865, 2012.

Skeffington, R. a., Whitehead, P. G., Heywood, E., Hall, J. R., Wadsworth, R. a. and Reynolds, B.: Estimating uncertainty in terrestrial critical loads and their exceedances at four sites in the UK, Sci. Total Environ., 382(2–3), 199–213, doi:10.1016/j.scitotenv.2007.05.001, 2007.

---

## Referee Report (RR1)

The authors have chosen to respond to most of the original comments in the online discussion rather than modifying the manuscript. Many of these responses are acceptable, but there are however a couple of important issues which have not been adequately addressed. Those original comments, along with the authors' response and a corresponding assessment are given below.

**Original comment:** P6,L19 Could the authors include their reasoning for choosing only 84 design points. The normal recommended minimum number for constructing Gaussian process emulators is 10 per input variable, which in this case would be 130. See, for example Loeppky et al, 2009, Choosing the Sample Size of a Computer Experiment: A Practical Guide, Technometrics.

**Authors' response:** We were aware of this paper cited by the reviewer. The paper states that an empirical $n = 10d$ rule (where $n$ is the sample size and $d$ is the number of variables under investigation) provides reasonable accuracy when approximating model response with a Gaussian process emulator. This rule is applicable to the model response of any complexity. The paper indicates this is an empirical rule and does not state that $10d$ is the minimum sample size for a Gaussian process emulator to perform well. In our particular study, the input-output response function is expected to be smooth, hence there is no need for a very large number of sample points as it will not reduce error of the emulator.
Again, it is a balance on return for computational resource. Ideally, in the situation where model runs are computationally expensive a sequential sampling technique can be applied to track the improvement of emulator performance with increase in the sample size.

**Assessment of response:** It is correct that the paper does not state that 10d model runs must be performed in every case, rather that it is a rule of thumb for an initial experiment, after which more runs may be performed if the emulators do not prove to be accurate enough. There are, however, published studies using fewer than 10d model runs, which I am sure the authors will be familiar with, and this is perfectly acceptable as long as a thorough emulator validation is performed – please see the point below.

**Original comment:** P7,L10 The authors state that the emulator error was estimated using cross validation and this is presented in the SI. However very little detail is given there except a reference to a paper describing the Matlab package used to construct the emulators (Lataniotis, 2017), which says that the package uses cross validation for parameter estimation, not emulator validation. The authors also say in the paragraph above that they used cross validation for parameter estimation. A clear statement is required as to whether or not the same cross validation was used for both parameter estimation and emulator validation. The accuracy of the emulators is of such key importance to everything that follows that summary statistics of either a separate cross validation, or a validation with a held-out data set, must be presented in the main text.

**Authors' response:** We have checked the cross-validation error values by performing a separate cross validation calculation. The results of explicitly-performed cross-validation do not differ from that which was originally presented in the supplementary material. We accept that the reference we originally cited may have caused some confusion, as the formula for cross-validation is reported in the parameter estimation section. The same formula was used for calculating the cross-validation error for the emulator. We now instead explicitly state in

the supplementary information the cross-validation equation used. The three figures in the supplementary information have now been replaced with the updated cross-validation versions. The code written to perform these independent cross-validations is also added to the data repository for this paper.

**Assessment of response:** Unfortunately the authors have chosen not to present the results of their emulator validation in the main text, as requested, despite this being one of the most critical parts of the analysis. Every other published study using these methods that I am aware of presents the emulator validation as an important part of the main text because the validity of all of the results of such studies is dependent upon it. The authors have instead made a small modification to the supporting information, but still only give the emulator error as a fraction of the model output variance, so that the reader has no way of knowing what the absolute magnitude of the emulation errors might be.
What has been presented in the supporting information would be a useful addition to a proper validation exercise, if the emulator variance were given as a fraction of the model output variance, rather than the emulator error, as this would support the use of the emulator mean value in the sensitivity calculations which follow.

**Original comment:** P14,L25 Given the authors assertion in the previous paragraph that the uncertainty in model output is likely to be driven mainly by variables that they have not included in their analysis, would they concede that their uncertainty estimates are likely to underestimate by a large degree the real uncertainty in the model output, i.e. that caused by uncertainty in all of the input variables plus the model discrepancy.

**Authors' Response:** Yes, we agree that the total uncertainty in the estimated concentration of the pollutants is higher than the uncertainty propagated from the input emissions only as there are other uncertain model inputs and parameters. Ideally sensitivity analysis should be incorporated as a part of the model development process (which could aid in both model simplification/reduction and calibration). In that approach the effect of all uncertain inputs and parameters could be assessed without having to do it retrospectively. In addition, screening techniques, e.g. the Morris method (Morris, 1991), could be applied to identify the inputs and parameters that most drive variation in the model outputs and which need to be investigated further. However, here we concentrate on presenting the application of the method itself and on a subset of inputs which previously was shown to drive uncertainty in the model output values.

**Assessment of response:** Given that the authors agree with this comment, it would seem appropriate to modify the manuscript accordingly.

---

## Author Response (AR2)

**Response to second review comments**
**acp-2018-690: Advanced methods for uncertainty assessment and global sensitivity analysis of a Eulerian atmospheric chemistry transport model**
by Aleksankina et al.

Comments made on our revised paper by the co-editor and the reviewer are given in italics.

**Author responses to Co-editor's non-public comments to the authors**
*I consulted with Reviewer #1 who opines that his/her comments have led to only a few minor revisions in the main text of the manuscript itself. Please consider whether revisions to your manuscript would be appropriate in response to Reviewer #1 comments #3, 4, 5, 8, 10, 11, 12, and 13. You have been much more responsive to comments from Reviewer #2 in terms of making revisions to the manuscript. With respect to comments from Reviewer #2, the addition of a qualifier ("so-called" compensation of errors) in response to comment #9 is not sufficient or helpful to readers. Please also consider whether revisions to the manuscript are warranted in response to Reviewer #2, comment #10.*

*I am fine with the meta-model evaluation being presented in supporting information as you have done if that is your preference.*

**Author response: The following revisions to the manuscript have been made in response to the comments 3, 4, 5, 8, 10, 11, 12, and 13 made by Reviewer #1 in their first review.**

Relating to Reviewer #1's comment no. 3: The following text has now been added to Section 2.2: "In this study, emissions of all of the major primary anthropogenic pollutant compounds were investigated. The decision to concentrate on the anthropogenic emissions was made based on the fact that one of the main applications of the EMEP4UK model is in providing scientific support for policy-making regarding impacts of interventions leading to anthropogenic emissions reductions. Hence the potential future changes in emissions driven by environmental and climate change policies are not likely to affect biogenic emissions as much as anthropogenic emissions. Therefore, it was decided to investigate model response to the changes in anthropogenic emissions."

Relating to Reviewer #1's comment no. 4: The following text has now been added to Section 2.2: "The shipping emissions were not split by pollutant because the inclusion of this split for this source comprised too great a computational cost for the analyses. Most shipping emissions do not impact on land-based population and ecosystem exposure, which was our focus, compared with the terrestrial emissions."

Relating to Reviewer #1's comment no. 5: The following text has now been added to Section 2.3: "In this study, 84 sampling points were found to be sufficient to create an adequately performing emulator as the input-output response function for the EMEP4UK model was expected to be smooth on monthly and annually averaged timescales. More generally, however, the case where model runs are computationally expensive and input-output relationship is less predictable a sequential sampling technique can be applied to track the improvement of emulator performance with increase in the sample size."

Relating to Reviewer #1's comment no. 8: As we noted in our original response, we disagree that our results show no difference between the London Marylebone Rd and London N. Kensington grid cells. We described and discussed the differences in the 'Results and discussions' section. On the matter of grid-averaging and 'site type', it is not within the scope of this paper to start discussion on issues associated with comparing model volume outputs with point source measurements; however, in order to emphasise that the names of 'urban background' and 'roadside' are not our designations but those assigned by the measurement network operators we have now amended text at the start of Section 3.3 to read: "for five different grid cells that were assigned the following environment types based on the site-type attributed to the national-network monitoring site…"

Relating to Reviewer #1's comment no. 10: The text now added to Section 2.2 in response to this reviewer's comment no. 5 (see above) also covers this comment, as there are no expected discontinuities or sharp peaks and troughs in the $O_3$ response to the input emissions on the annual or monthly timescale.

Relating to Reviewer #1's comment no. 11: We do not believe there is any additional benefit to our paper of adding a statement somewhere that the annual average 8-hour maximum $O_3$ concentration metric is well correlated to the annual average concentration metric. We have responded to the reviewer's question on this in our online responses to the comment when it was originally made.

Relating to Reviewer #1's comment no. 12: Our text already includes a statement that the surface concentrations of the modelled pollutants in the UK may be dominated by the precursor emissions and long-range transport from outside the UK and are therefore relatively insensitive to changes in the UK emissions. We have now also added additional text to the paper relating to the reviewer's comments nos. 3 (as above) and 13 (as below) that further acknowledges that our work focused only on investigations of model uncertainty via land-based emissions.

Relating to Reviewer #1's comment no. 13: The following text has now been added to Section 3.4: "Finally, in this study the overall model output uncertainty is likely to be lower than the theoretical total model output uncertainty, as in addition to the input emissions there is a variety of other uncertain model inputs. Assessing the effect of variation in every model input and parameter on the model output is a laborious task, hence ideally sensitivity analysis should be incorporated as a part of the model development process. By using this approach, the effect of all uncertain inputs and parameters could be assessed without having to do it retrospectively."

**Author response: The following revisions to the manuscript have been made in response to the comments no. 9 and 10 made by Reviewer #2 in their first review.**

Relating to Reviewer #2's comment no. *9:*
***Original comment*** *Results Section 3.1: You refer to the 'compensation of errors' as one explanation why the surface response is weak given the input uncertainties. Can you point to the literature for evidence of this statement? I've only seen "compensation of errors" only referred to in the context of process representation in models.*
***Our original Response****: The phrase comes from Skeffington et al. 2007. In that paper the reason for narrowing of confidence limits for critical loads compared to those of the input*

*parameters was explained to be due to a "compensation of errors" mechanism, but no further explanation was provided. Here, by compensation of errors we mean a situation when the variation in the output is less than expected. This could be caused by multiple inputs having an opposite effect on the magnitude of change in the output of interest.*
*We have changed the phrasing in our paper to "so-called compensation of errors".*

**Updated response:** Perhaps the co-editor did not notice that the text in the manuscript already explains what was meant by 'compensation error', viz: "Another explanation is the 'so-called compensation of errors' whereby a positive effect of one or multiple input variables on the output is compensated by a negative effect of another input variable(s). This leads to the narrower confidence intervals associated with the EMEP4UK outputs."
At the last paper revision stage we chose to add the qualifier 'so called' so as to emphasise that we were using the 'compensation error' terminology used by Skeffington et al. (2007) and that it was not our phrasing. We have now added the Skeffington et al. (2007) reference to the relevant sentence in the paper.

Relating to Reviewer #2's comment no. *10:*
***Original comment*** *Results Section 3.3: One potential explanation for the seasonal change in sensitivity at Harwell to shipping emissions is the seasonal change in the wind direction which results in more NOx from shipping emissions being transported to the site. Can this be verified from the WRF meteorology used to drive the model?*
***Our original response:*** *The seasonal wind speed and direction for the year 2012 is shown in Figure A below, using the meteorology supplied from the AURN data as extracted using the openair package. It could be argued that there is some correlation between sensitivity index patterns in Figure 8 of our paper and the wind direction; however most likely the seasonality in NOx sensitivity to shipping emissions is due to a combination of interacting processes within the model.*

**Updated response:**
The wind rose for Harwell AURN site for year 2012 has been added to the supplementary information and has been referenced in the main text of the manuscript in Section 3.3.
"Potential explanation for this is seasonal change in the wind direction which results in more $NO_x$ from shipping emissions being transported to the grid cell during the summer months (the wind rose is presented in Fig. S3)."

**Author responses to Reviewer #1's second review report**

*The authors have chosen to respond to most of the original comments in the online discussion rather than modifying the manuscript. Many of these responses are acceptable, but there are however a couple of important issues which have not been adequately addressed. Those original comments, along with the authors' response and a corresponding assessment are given below.*

*__Original comment:__ P6,L19 Could the authors include their reasoning for choosing only 84 design points. The normal recommended minimum number for constructing Gaussian process emulators is 10 per input variable, which in this case would be 130. See, for example Loeppky et al, 2009, Choosing the Sample Size of a Computer Experiment: A Practical Guide, Technometrics.*

*__Authors' response:__ We were aware of this paper cited by the reviewer. The paper states that an empirical n = 10d rule (where n is the sample size and d is the number of variables under investigation) provides reasonable accuracy when approximating model response with a Gaussian process emulator. This rule is applicable to the model response of any complexity. The paper indicates this is an empirical rule and does not state that 10d is the minimum sample size for a Gaussian process emulator to perform well. In our particular study, the input-output response function is expected to be smooth, hence there is no need for a very large number of sample points as it will not reduce error of the emulator. Again, it is a balance on return for computational resource. Ideally, in the situation where model runs are computationally expensive a sequential sampling technique can be applied to track the improvement of emulator performance with increase in the sample size.*

*__Assessment of response:__ It is correct that the paper does not state that 10d model runs must be performed in every case, rather that it is a rule of thumb for an initial experiment, after which more runs may be performed if the emulators do not prove to be accurate enough. There are, however, published studies using fewer than 10d model runs, which I am sure the authors will be familiar with, and this is perfectly acceptable as long as a thorough emulator validation is performed – please see the point below.*

Authors' response: We don't think that any further comment is needed here, nor any further modification to our paper. The reviewer is agreeing with us that 10d sample points is a rule of thumb given by one paper and that other studies have used fewer – we respond to the reviewer's comment on emulator validation below.

*__Original comment:__ P7,L10 The authors state that the emulator error was estimated using cross validation and this is presented in the SI. However very little detail is given there except a reference to a paper describing the Matlab package used to construct the emulators (Lataniotis, 2017), which says that the package uses cross validation for parameter estimation, not emulator validation. The authors also say in the paragraph above that they used cross validation for parameter estimation. A clear statement is required as to whether or not the same cross validation was used for both parameter estimation and emulator validation. The accuracy of the emulators is of such key importance to everything that follows that summary statistics of either a separate cross validation, or a validation with a held-out data set, must be presented in the main text.*

*__Authors' response:__ We have checked the cross-validation error values by performing a separate cross validation calculation. The results of explicitly-performed cross-validation do not differ from that which was originally presented in the supplementary material. We accept that the reference we originally cited may have caused some confusion, as the formula for cross-validation is reported in the parameter estimation section. The same formula was used*

*for calculating the cross-validation error for the emulator. We now instead explicitly state in the supplementary information the cross-validation equation used. The three figures in the supplementary information have now been replaced with the updated cross-validation versions. The code written to perform these independent cross-validations is also added to the data repository for this paper.*

***Assessment of response:*** *Unfortunately the authors have chosen not to present the results of their emulator validation in the main text, as requested, despite this being one of the most critical parts of the analysis. Every other published study using these methods that I am aware of presents the emulator validation as an important part of the main text because the validity of all of the results of such studies is dependent upon it. The authors have instead made a small modification to the supporting information, but still only give the emulator error as a fraction of the model output variance, so that the reader has no way of knowing what the absolute magnitude of the emulation errors might be.*

*What has been presented in the supporting information would be a useful addition to a proper validation exercise, if the emulator variance were given as a fraction of the model output variance, rather than the emulator error, as this would support the use of the emulator mean value in the sensitivity calculations which follow.*

Authors' response: As stated in our paper, k-fold cross-validation has been used to evaluate the performance of the chosen emulator; the results of the cross-validation and the code used to perform it are made available. We chose not to overwhelm the manuscript and the reader with extensive detail of the code implementation of the Gaussian Process emulator and its validation as the original MATLAB code for the emulator is a part of a freely available package UQLab. Additionally, the manuscript concentrates on the application of already developed methods (emulation, sensitivity analysis, and uncertainty analysis) rather than the development of those numerical methods.

The co-editor has confirmed that they are happy that our evaluation of the emulator is presented in the supporting information.

We have chosen to present the result of k-fold cross-validation (CV) as a pollutant concentration value (i.e. value of the model output) divided by the overall variance of the model output in order to make the number more meaningful. In the areas of the modelled domain where model is not sensitive to changes in the input emissions the variance around the predicted mean is very small, hence the CV error is going to be small. By presenting CV errors relative to the variance of the output for the corresponding grid cell, we avoid falling into a trap of stating that in these grid cells the emulator performs well because we take into account the sensitivity to the underlying inputs.

***Original comment:*** *P14,L25 Given the authors assertion in the previous paragraph that the uncertainty in model output is likely to be driven mainly by variables that they have not included in their analysis, would they concede that their uncertainty estimates are likely to underestimate by a large degree the real uncertainty in the model output, i.e. that caused by uncertainty in all of the input variables plus the model discrepancy.*

***Authors' Response:*** *Yes, we agree that the total uncertainty in the estimated concentration of the pollutants is higher than the uncertainty propagated from the input emissions only as there are other uncertain model inputs and parameters. Ideally sensitivity analysis should be incorporated as a part of the model development process (which could aid in both model*

*simplification/reduction and calibration). In that approach the effect of all uncertain inputs and parameters could be assessed without having to do it retrospectively. In addition, screening techniques, e.g. the Morris method (Morris, 1991), could be applied to identify the inputs and parameters that most drive variation in the model outputs and which need to be investigated further. However, here we concentrate on presenting the application of the method itself and on a subset of inputs which previously was shown to drive uncertainty in the model output values.*

***Assessment of response:*** *Given that the authors agree with this comment, it would seem appropriate to modify the manuscript accordingly.*

Authors's response: In addition to the existing manuscript text: "[Overall uncertainty was found to be low]. This indicates that the variation in the input data (i.e. emissions) does not cause a substantial variation in the outputs. Our results indicate, that this can likely be explained by variations in the other model input parameters such as chemical reaction rates, deposition velocities or physical constant values which might cause more variation in the model outputs. Alternatively, surface concentrations of the modelled pollutants in the UK may be dominated by the precursor emissions and long-range transport from outside the UK and are therefore relatively insensitive to changes in the UK emissions."
The following text has now been added to the end of Section 3.4: "Finally, in this study the overall model output uncertainty is likely to be lower than the theoretical total model output uncertainty, as in addition to the input emissions there is a variety of other uncertain model inputs. Assessing the effect of variation in every model input and parameter on the model output is a laborious task, hence ideally sensitivity analysis should be incorporated as a part of the model development process. By using this approach, the effect of all uncertain inputs and parameters could be assessed without having to do it retrospectively."